# A Ubl/ubiquitin switch in the activation of Parkin

Véronique Sauvé[1,†], Asparouh Lilov[1,†], Marjan Seirafi[1,†], Marta Vranas[2], Shafqat Rasool[1,2], Guennadi Kozlov[1], Tara Sprules[3], Jimin Wang[4], Jean-François Trempe[2,*] & Kalle Gehring[1,**]

## Abstract

**Mutations in Parkin and PINK1 cause an inherited early-onset form of Parkinson's disease. The two proteins function together in a mitochondrial quality control pathway whereby PINK1 accumulates on damaged mitochondria and activates Parkin to induce mitophagy. How PINK1 kinase activity releases the auto-inhibited ubiquitin ligase activity of Parkin remains unclear. Here, we identify a binding switch between phospho-ubiquitin (pUb) and the ubiquitin-like domain (Ubl) of Parkin as a key element. By mutagenesis and SAXS, we show that pUb binds to RING1 of Parkin at a site formed by His302 and Arg305. pUb binding promotes disengagement of the Ubl from RING1 and subsequent Parkin phosphorylation. A crystal structure of Parkin Δ86–130 at 2.54 Å resolution allowed the design of mutations that specifically release the Ubl domain from RING1. These mutations mimic pUb binding and promote Parkin phosphorylation. Measurements of the E2 ubiquitin-conjugating enzyme UbcH7 binding to Parkin and Parkin E3 ligase activity suggest that Parkin phosphorylation regulates E3 ligase activity downstream of pUb binding.**

**Keywords** mitochondria; mitophagy; Parkinson's disease; phosphorylation; ubiquitination

**Subject Categories** Molecular Biology of Disease; Post-translational Modifications, Proteolysis & Proteomics; Structural Biology

The EMBO Journal (2015) 34: 2492–2505

## Introduction

Parkinson's disease is one of the most common neurodegenerative diseases. Its characteristic motor symptoms are caused by the loss of dopaminergic neurons in the *substantia nigra* area of the midbrain. While most cases are sporadic and occur later in life, about 5–10% of the cases are caused by somatic genetic mutations (Martin *et al*, 2011). Among those, mutations in *PARK2* (Parkin) and *PARK6*

(PINK1) are responsible for the majority of early-onset recessive occurrences of the disease (Kitada *et al*, 1998; Valente *et al*, 2004).

Numerous studies over the last 10 years have implicated Parkin and PINK1 in a common mitochondrial quality control pathway (Narendra *et al*, 2012). In *Drosophila*, loss of PINK1 causes mitochondrial dysfunction that can be complemented by Parkin (Clark *et al*, 2006; Park *et al*, 2006). Parkin is a cytosolic E3 ubiquitin ligase that is recruited to mitochondria damaged by depolarization, reactive oxygen species (ROS), or accumulation of unfolded proteins (Narendra *et al*, 2008; Jin & Youle, 2013; Ashrafi *et al*, 2014). Parkin then ubiquitinates mitochondrial outer membrane proteins such as Mitofusin or Miro to induce a wide range of outcomes, from proteasomal degradation to vesicle formation, motility arrest, and mitochondrial autophagy (Narendra *et al*, 2008; Gegg *et al*, 2010; Tanaka *et al*, 2010; Ziviani *et al*, 2010; Wang *et al*, 2011; McLelland *et al*, 2014). Critically, all these Parkin-dependent quality control processes require PINK1 (PTEN-induced putative kinase 1), a Ser/Thr kinase with a mitochondrial targeting sequence. PINK1 is normally imported into polarized mitochondria where it is cleaved by proteases and further degraded by the proteasome (Jin *et al*, 2010; Deas *et al*, 2011; Meissner *et al*, 2011; Greene *et al*, 2012). When mitochondria are damaged, for example, upon depolarization, PINK1 accumulates on the cytosolic face of the outer membrane (Zhou *et al*, 2008) where it can recruit and activate Parkin (Geisler *et al*, 2010; Matsuda *et al*, 2010; Narendra *et al*, 2010).

The kinase activity of PINK1 is directly involved in Parkin translocation and activation. First of all, PINK1 directly phosphorylates Ser65 located in the Ubl domain of Parkin (Kondapalli *et al*, 2012; Shiba-Fukushima *et al*, 2012). This leads to an increase in Parkin ligase activity. Secondly, PINK1 also phosphorylates ubiquitin on Ser65 (Kane *et al*, 2014; Kazlauskaite *et al*, 2014b; Koyano *et al*, 2014). Phosphorylated ubiquitin (pUb) interacts with Parkin and also enhances its activity. Moreover, phosphorylated ubiquitin can act as a receptor for Parkin translocation to mitochondria (Okatsu *et al*, 2015) in a feed-forward mechanism (Ordureau *et al*, 2014). Thus, PINK1 phosphorylation of both ubiquitin and the Ubl are required for Parkin's function in mitochondrial quality control, but their respective roles remain controversial.

Structural and biochemical studies have unveiled the molecular mechanisms underlying Parkin's E3 ubiquitin ligase activity. Parkin

1 Groupe de recherché axé sur la structure des protéines and Department of Biochemistry, McGill University, Montréal, QC, Canada
2 Department of Pharmacology and Therapeutics, McGill University, Montréal, QC, Canada
3 Quebec/Eastern Canada High Field NMR Facility (QANUC), Montréal, QC, Canada
4 Department of Molecular Biophysics and Biochemistry, Yale University, New Haven, CT, USA
 *Corresponding author. Tel: +1 514 398 6833; E-mail: jeanfrancois.trempe@mcgill.ca
 **Corresponding author. Tel: +1 514 398 7287; E-mail: kalle.gehring@mcgill.ca
 †These authors contributed equally to this work

is a member of the RBR (RING-in-between-RING) ligase family, which uses a RING/HECT hybrid mechanism to transfer ubiquitin from an E2 enzyme to a substrate via a thioester intermediate (Wenzel *et al*, 2011). Crystal structures of full-length Parkin (Trempe *et al*, 2013) and the C-terminal domains (Riley *et al*, 2013; Trempe *et al*, 2013; Wauer & Komander, 2013) showed that under basal conditions, the protein adopts a closed conformation where multiple intramolecular interactions inhibit key sites that are required for ubiquitin transfer. In particular, the E2-binding site on RING1 is occluded by the repressor element of Parkin (REP), and juxtaposed to the Ubl domain that was shown to play an important role in the regulation of its activity (Chaugule *et al*, 2011; Trempe *et al*, 2013). However, the structural changes induced by Ubl phosphorylation and pUb binding to Parkin, and the biochemical steps they regulate, remain obscure.

Here, we present a higher resolution crystal structure of Parkin with a deletion in the linker that follows the Ubl domain. The structure provides a detailed snapshot of the molecular interactions between the Ubl and RING1 domains. Using isothermal titration calorimetry (ITC), NMR, and small-angle X-ray scattering (SAXS), we show that phosphorylation of the Ubl domain releases its interaction with the Parkin RING1 domain and that this increases the affinity of pUb binding to Parkin. Conversely, pUb binding releases Ubl and facilitates its phosphorylation. Through screening of basic residues in Parkin, we identify His302 and Arg305 as essential residues for binding pUb. SAXS data modeling shows that the pUb-binding site is located on the opposite side of the Ubl-binding patch on RING1, suggesting that the antagonism between Ubl and pUb is mediated through negative allostery. We show that Parkin phosphorylation plays additional roles promoting UbcH7 binding and increasing Parkin E3 ligase activity. Our data are consistent with a model in which Parkin is initially recruited to mitochondria by binding to pUb. This releases the Ubl domain to promote Parkin phosphorylation, which is the downstream signal for releasing its ligase activity.

# Results

## Improved structure of Δ86–130 Parkin reveals how Ubl binds to RING1

We previously reported the low-resolution X-ray structure of full-length Parkin that showed Parkin adopts an auto-inhibited arrangement in which both the E2-binding site and catalytic site are obstructed by interdomain contacts (Trempe *et al*, 2013). Higher resolution structures of the R0RBR fragment of Parkin structure confirmed the auto-inhibition (Riley *et al*, 2013; Trempe *et al*, 2013; Wauer & Komander, 2013). Parkin consists of an N-terminal Ubl domain, followed by a long linker leading to four zinc-binding domains: RING0, RING1, IBR, and RING2, collectively referred to as the R0RBR module. In the full-length structure, the Ubl domain binds to RING1. No density was observed for the linker between the Ubl domain and the R0RBR module in full-length Parkin structure (Trempe *et al*, 2013), presumably due to its flexibility and mobility. We reasoned that the linker might be responsible of the low "quality" of the full-length Parkin crystals. The length of this linker varies significantly between different species, so we designed a version lacking its least conserved section spanning residues 86–130

(Δ86–130 Parkin; Fig 1A). This version of Parkin was crystallized and its structure was determined at a resolution of 2.54 Å (Fig 1B, Table 1).

The crystal packing gave rise to two types of contacts between the Ubl domain and the R0RBR module (Fig EV1A). In one contact, the Ubl domain interacts mainly with RING1 near the IBR domain as previously observed in our low-resolution structure of full-length Parkin (model 1, Fig EV1B). In the other, the Ubl stacks against the REP linker and the RING1 domain near the RING2 domain (model 2, Fig EV1B). Solution SAXS data acquired on Δ86–130 Parkin fitted much better to model 1 ($\chi^2 = 1.60$) than to model 2 ($\chi^2 = 7.60$), thus confirming that Ubl binds to the RING1 helix as previously reported (Fig EV1C).

The higher resolution Δ86–130 Parkin structure superimposes well on the full-length Parkin structure (backbone rmsd of 1.6 Å), showing that the reduced length of the linker does not perturb the conformation of Parkin (Fig EV1D). Intramolecular interactions between RING0 and RING2 block Cys431 in the catalytic site, and the E2-binding site is occluded by the Ubl domain and linker between the IBR and RING2 domains (Fig 1B). This linker contains a well-ordered helix, the REP, surrounded by flexible residues. In the new structure, residues 382–391 preceding the REP are again disordered, but electron density for the residues following the REP could be observed. This REP-RING2 linker adopts different conformations in different structures with high B-factors, which suggests that it is mainly structured by intermolecular contacts arising from crystal packing. We similarly observe flexibility of the IBR domain in the Δ86–130 Parkin structure. Unlike previous Parkin crystals (Riley *et al*, 2013; Trempe *et al*, 2013; Wauer & Komander, 2013), the IBR domain is not stabilized by crystal contacts with other molecules. The IBR electron density is thus poorly resolved in the Δ86–130 Parkin structure suggesting intrinsic flexibility, consistent with a previous NMR study (Beasley *et al*, 2007) and the wide range of positions adopted by the IBR among the different chains in the R0RBR structure (Fig EV1E) (Riley *et al*, 2013; Trempe *et al*, 2013; Wauer & Komander, 2013).

As observed in a previous structure (Wauer & Komander, 2013), the Δ86–130 Parkin crystal structure contains several bound sulfate ions. Two of the sites are conserved in both molecules in the asymmetric unit. The first and best-ordered site is on RING1 opposite the Ubl-binding surface and formed by Arg305 and Tyr312 (Fig 1C). The second site is at the interface of the RING0 and RING2 domains and formed by Lys161, Arg163, and Lys211 (Fig 1D).

The new structure reveals the specific interactions that dock the Ubl domain to the rest of Parkin. The Ubl domain mainly interacts with helix 261–274 of the RING1 domain (Fig 1E). Arg6 and His68 form hydrogen bonds with Asp274, a residue located at the end of the helix and preceding Arg275, a frequent mutation site in familial PD (R275W). Like ubiquitin, Ubl has a hydrophobic patch centered on Ile44, which stacks against Leu266 of RING1 helix 261–274 (Fig 1E). Asn8, a residue conserved in Parkin orthologs, binds RING1 helix 309–316 via a hydrogen bond with Gln311 and Arg314 (Fig 1F). This position is leucine in ubiquitin and may be a specificity determinant as ubiquitin does not bind the R0RBR module. Ser65 of the Ubl domain, which is phosphorylated by PINK1 (Shiba-Fukushima *et al*, 2012; Kazlauskaite *et al*, 2014a), faces the disordered residues 382–391 preceding the REP helix and is located within only 12 Å from RING1 (Fig 1B).

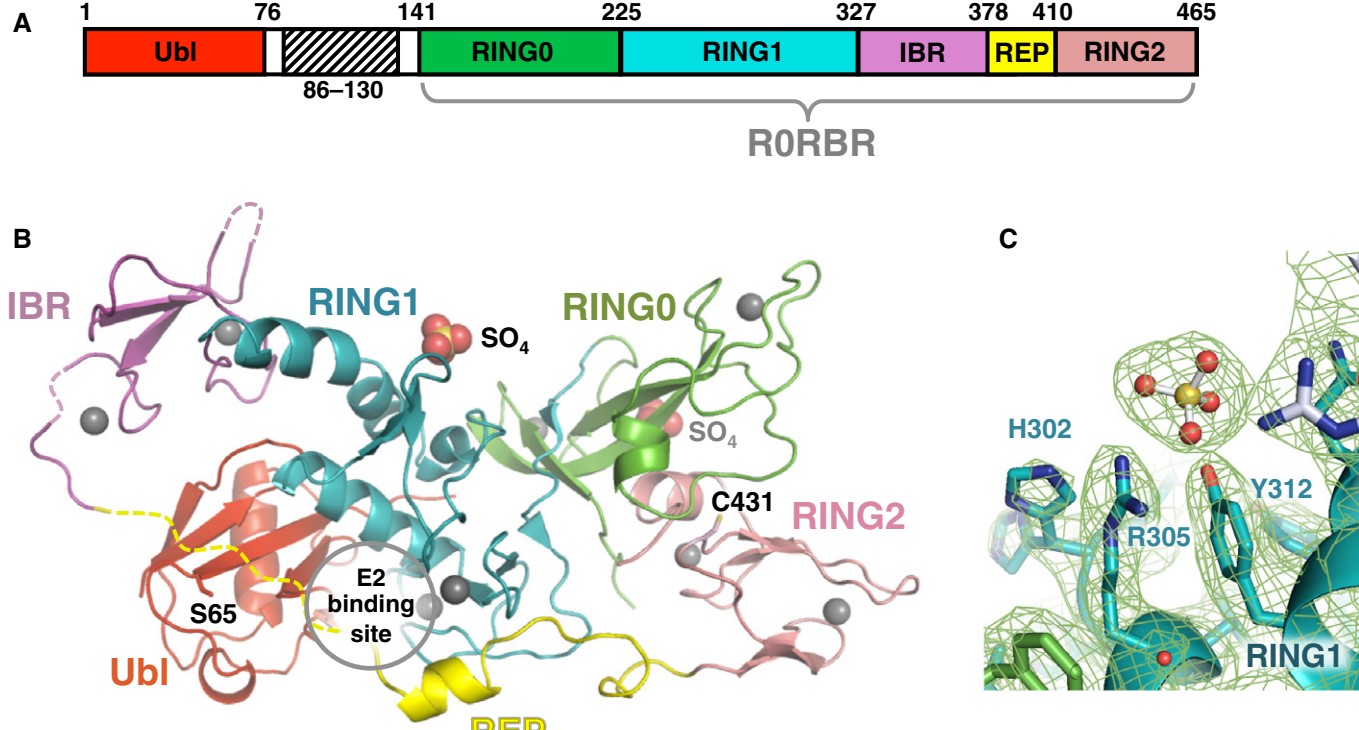

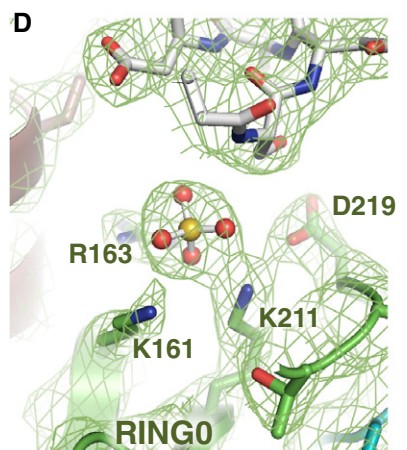

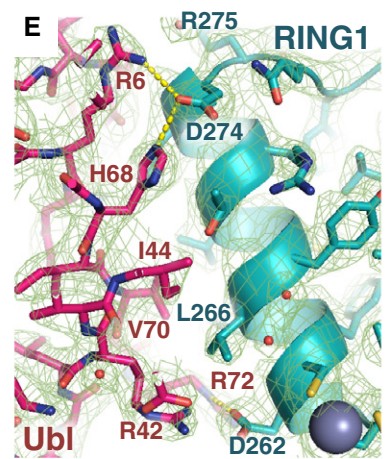

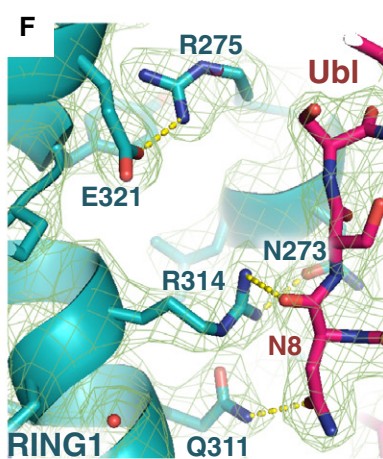

**Figure 1. Structure of ∆86–130 Parkin reveals details of Ubl and sulfate ion binding.**

A Domain organization of Parkin, showing location of ∆86–130 deletion and definition of R0RBR module.

B Structure of ∆86–130 Parkin. The sites of phosphorylation on Ser65 of the Ubl domain and the catalytic residue Cys431 are indicated along with the bound sulfates.

C, D Polar contacts and residues around the bound sulfate ions. The sulfate in the first site has B-factor of 40 Å$^2$ and the second has a B-factor of 80 Å$^2$.

E, F Close-up views of the interaction between the Ubl (red) and RING1 domains (cyan).

### Parkin Ser65 phosphorylation releases Ubl from RING1

The proximity of Ser65 to RING1 prompted us to test the role of its phosphorylation in regulating the Ubl–RING1 interaction. We used ITC to probe the interaction between the R0RBR module and PINK1-phosphorylated Ubl (pUbl). A $K_d$ of 16 µM was obtained for the unmodified Ubl binding to R0RBR (Figs 2A and EV2A), in agreement

with previous measurements (Chaugule *et al*, 2011). Strikingly, no signal was detected for the titration of R0RBR with pUbl. To confirm that the lack of ITC signal was due to a loss of interaction and not the absence of an enthalpy change (∆H), we performed an NMR titration experiment. As previously reported, $^{15}$N-Ubl signals disappeared upon addition of R0RBR due to the formation of a high molecular weight complex (Fig 2B) (Chaugule *et al*, 2011). Addition

**Table 1.** Data collection and refinement statistics for Δ86–130 Parkin.

| Data collection | |
|---|---|
| Space group | H3 |
| Cell dimensions | |
| a,b,c (Å) | 114.11, 114.11, 186.40 |
| α, β, γ (°) | 90.0, 90.0, 120.0 |
| Resolution (Å) | 36.62–2.54 (2.63–2.54)[a] |
| Pearson's CC* | 0.943 (0.469) |
| $R_{merge}$ | 0.128 (> 1.000) |
| $<I/\sigma(I)>$ | 5.1 (1.8) |
| Completeness (%) | 98.98 (92.18) |
| Redundancy | 8.1 (2.8) |
| **Refinement** | |
| Resolution (Å) | 36.62–2.54 |
| Used reflections | 29,521 |
| $R_{work}/R_{free}$ | 0.193/0.235 |
| $CC_{work}/CC_{free}$ | 0.945/0.918 |
| Number of atoms | |
| Protein | 5714 |
| $Zn^{2+}$ | 16 |
| Water | 64 |
| Sulfate | 30 |
| B-factors | |
| Protein/$Zn^{2+}$ | 70.5 |
| Water | 36.6 |
| Sulfate | 91.9 |
| R.m.s. deviations | |
| Bond lengths (Å) | 0.009 |
| Bond angles (°) | 1.35 |
| Ramachandran statistics | |
| Favored region (%) | 95 |
| Allowed region (%) | 5 |
| Disallowed region (%) | 0 |

[a]Data in the highest resolution shell shown in parentheses.

of R0RBR to the phosphorylated Ubl domain led to only a small degree of signal loss consistent with a weaker interaction (Fig 2C). This was further confirmed with a pull-down assay that showed that Ubl, but not pUbl, bound to immobilized GST-R0RBR (Fig EV2B). All three techniques indicate that Ubl phosphorylation strongly decreases binding to the Parkin R0RBR module.

To confirm that the Ubl domain was specifically binding to RING1, we used our crystal structure of Δ86–130 Parkin to design a mutation that disrupts the Ubl–RING1 interface. Leu266, which interacts with the hydrophobic patch on the Ubl (Fig 1E), was mutated to a lysine to create a repulsive interaction. No binding of the isolated Ubl domain to L266K R0RBR could be observed by ITC or NMR (Fig 2A and B).

We next asked whether Ser65 phosphorylation induces the dissociation of the Ubl domain in the context of the full-length protein.

Dissociation of intramolecular interactions should increase the radius of gyration ($R_g$) and increase the number of long interatomic distances in the structure. SAXS measurements of unmodified full-length Parkin yielded an $R_g$ of 29.1 Å and a molecular weight estimations near its 52-kDa monomeric mass (Figs 2D and EV2C). SAXS analysis of phosphorylated Parkin (pParkin) showed a significantly greater $R_g$ of 31.9 Å. The increased $R_g$ was not caused by aggregation, because the estimated mass of the particle remained the same. The L266K Parkin mutant also showed a larger $R_g$ without evidence of aggregation. The $P(r)$ pair-distance distribution functions calculated from the three samples confirmed more extended distances ($r > 80$ Å) in pParkin and L266K Parkin (Fig EV2E). Changes in Parkin upon phosphorylation were also observed during sample preparation. We detected a significantly earlier elution of pParkin upon size-exclusion chromatography, suggesting an increased hydrodynamic size (Fig EV2D). The L266K Parkin mutant and a second mutation, N273K, that disrupts the Ubl–RING1 interface similarly eluted earlier than wild-type Parkin (Fig EV2D). In agreement with the experiments with the isolated Ubl domain, these measurements show that phosphorylation of Parkin induces a structural rearrangement consistent with dissociation of the Ubl domain from RING1.

We next performed NMR experiments to check whether the loss of binding induced by Ser65 phosphorylation could be attributed to an alternative conformation of the Ubl domain, as observed for pUb (Wauer *et al*, 2015b). In the latter case, the C-terminal β5-strand of pUb was shown to adopt two different configurations in solution: a major conformation similar to unmodified ubiquitin and a minor conformation, unique to pUb, in which the β5-strand was shifted by two residues. We measured $^{15}$N-$^1$H chemical shift differences between Ubl and pUbl (Fig EV2F and G), and those perturbations were mapped on the Ubl domain of the structure of Parkin (Fig 2E). The largest shifts were located in the region around Ser65: β-strand 65–70, its preceding loop 60–65, and its adjacent β-strand 2–6. All those regions face RING1 or the linker connecting the IBR to the REP. Interestingly, the backbone amide of Asn8, a residue unique to Parkin and involved in the Ubl–RING1 interaction, is one of the most disturbed resonances despite being located 17 Å away from Ser65. Ubl phosphorylation thus seems to induce far-reaching perturbations in the structure of the Ubl. However, comparison of $^{15}$N-NOESY-HSQC experiments acquired on Ubl and pUbl showed no major differences in cross-strands NOE patterns (Fig EV2H), suggesting that no shift in β-strand alignment occurs as observed in pUb (Wauer *et al*, 2015b). The large perturbations in the amide chemical shifts probably reflect reconfiguration of the side chains around the phosphorylation site. These different side-chain conformations affect surface residues further from the phosphorylation site such as Asn8 that in turn affect the interaction with RING1.

## The Ubl domain and pUb compete for binding to Parkin

Previous studies have demonstrated that the addition of pUb to Parkin enhances ubiquitin chain formation and E2 ubiquitin discharging (Kane *et al*, 2014; Kazlauskaite *et al*, 2014b; Koyano *et al*, 2014); however, the molecular details of the interaction are unknown. We thus examined binding of different Parkin constructs to pUb by ITC. We measured a 20-fold higher affinity for pUb binding to R0RBR (22 nM) than to full-length Parkin (431 nM) (Figs 3A

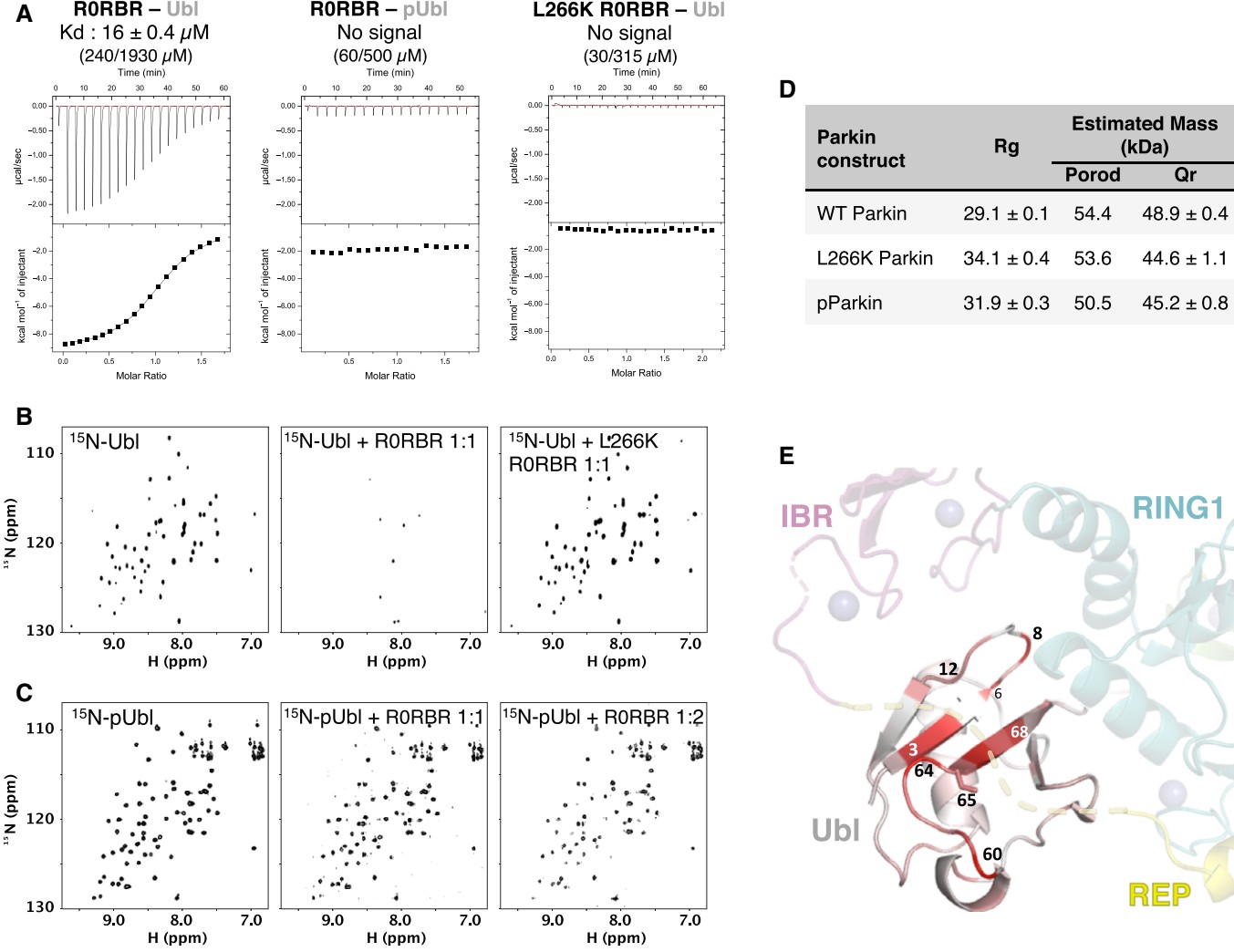

**Figure 2. Phosphorylation of Parkin disrupts the Ubl–RING1 interaction.**

A   ITC measurement of the isolated Parkin Ubl or phosphorylated Ubl domain with wild-type (WT) or L266K Parkin R0RBR fragments. The protein concentrations in the cell and syringe are indicated in parentheses.

B   NMR $^1$H-$^{15}$N correlation spectra of unmodified $^{15}$N-Ubl (100 μM) titrated with wild-type or mutant L266K R0RBR fragments. Binding results in a loss of NMR signals.

C   Spectra of phosphorylated $^{15}$N-pUbl (30 μM) titrated with the Parkin R0RBR fragment.

D   SAXS-derived parameters for different full-length Parkin constructs.

E   Backbone amide NMR chemical shifts perturbations induced by Ser65 phosphorylation in the Ubl domain of Parkin. The magnitude of the perturbation is indicated by the intensity of the red color displayed on Ubl bound to RING1.

and EV3A). This latter affinity is similar to that reported by another group using ITC and biolayer interferometry (Ordureau *et al*, 2014). Therefore, deletion of the Ubl increases the affinity of pUb for Parkin.

To test whether the inhibitory role of the Ubl was linked to its ability to bind RING1, we tested pUb binding to the L266K mutant. ITC measurements showed that full-length L266K Parkin binds pUb with an affinity comparable to the R0RBR module (37 versus 22 nM), confirming that the Ubl–RING1 interaction interferes with pUb binding (Fig 3A). Similar results were obtained by mutating Asn273, which is also located at the Ubl–RING1 interface and required for Ubl binding. In the absence of the Ubl domain, the L266K mutation did not further increase the affinity of pUb binding, implying that the enhanced affinity observed with the L266K and

N273K mutants was linked to the release of the Ubl domain (Figs 3A and EV3A).

We next investigated the effect of the phosphorylation of Parkin Ubl on pUb binding since it has already been demonstrated that both pUbl and pUb were required for optimal Parkin activity (Kane *et al*, 2014; Kazlauskaite *et al*, 2014b; Koyano *et al*, 2014). Moreover, we have shown here that Parkin phosphorylation releases the Ubl (Fig 2). We confirmed by ITC that pParkin binds pUb better than unphosphorylated Parkin (Ordureau *et al*, 2014) (Fig 3A). We obtained a $K_d$ of 17 nM for pParkin, which contrasts with the 431 nM found for unmodified Parkin, but is similar to that measured for R0RBR and L266K Parkin. Parkin Ubl phosphorylation enhanced the binding of pUb to Parkin to the same extent as deletion of the Ubl domain or release of the Ubl domain from its binding

A

| Parkin construct | Affinity for pUb (nM) |
|---|---|
| WT Parkin | 431 ± 25 |
| R0RBR | 22 ± 5 |
| L266K Parkin | 37 ± 7 |
| L266K R0RBR | 41 ± 14 |
| N273K Parkin | 57 ± 17 |
| pParkin | 17 ± 3 |

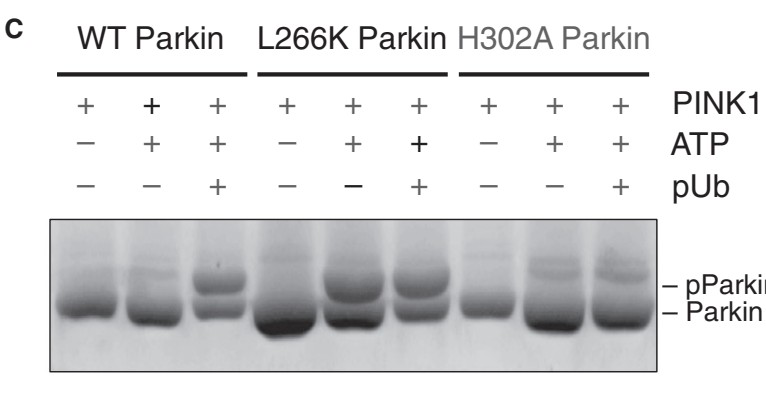

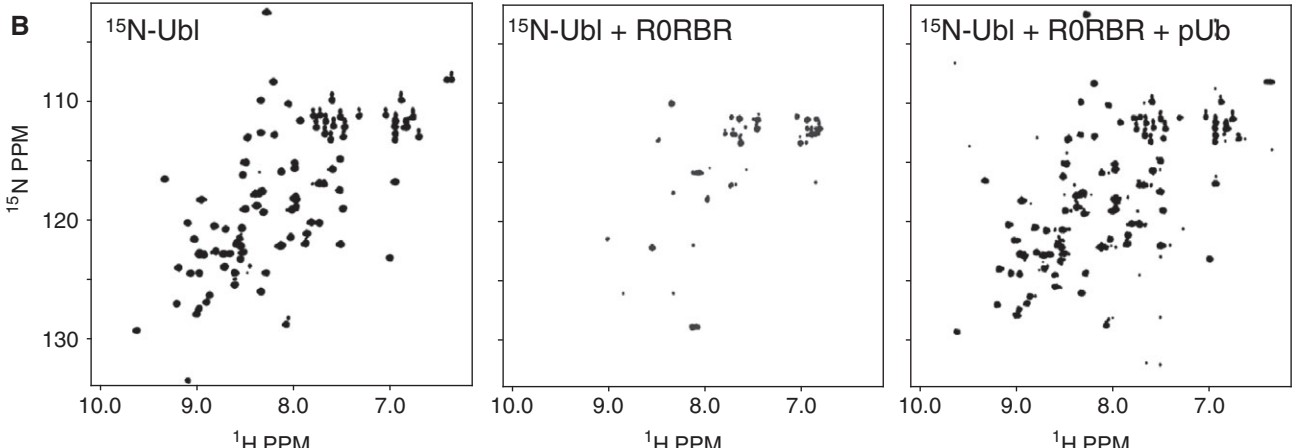

**Figure 3.  The Ubl domain and pUb compete for binding to Parkin.**

A   ITC-derived affinities of pUb binding to different Parkin constructs.
B   NMR competition experiment. $^{1}$H-$^{15}$N correlation spectra were acquired for $^{15}$N-Ubl (100 µM) in the presence of equimolar Parkin R0RBR and after the addition of a molar equivalent of pUb.
C   Parkin phosphorylation assay showing increased phosphorylation in the presence of pUb. Wild-type (WT), L266K, and H302A Parkin were incubated for 5 min with the TcPINK1 kinase with and without ATP and pUb. Products were resolved by Phos-tag SDS–PAGE and stained with Coomassie Blue.

site on RING1. This confirms that the Ubl phosphorylation weakens the Ubl–R0RBR interaction and facilitates access of pUb to Parkin.

The competition between pUb and Ubl binding to Parkin was tested by NMR spectroscopy. The loss of NMR signals from $^{15}$N-Ubl in the presence of the R0RBR module could be reversed by the subsequent addition of pUb (Fig 3B). This indicates that pUb was able to displace the Ubl domain from the R0RBR module.

PINK1 increases Parkin activity by phosphorylating Ser65 in the Ubl domain (Kondapalli *et al*, 2012). In the basal state, the Ubl domain is bound to RING1 and Ser65 is poorly accessible for modification. We hypothesized that the release of Ubl was necessary for efficient phosphorylation of Ser65 by PINK1. We thus measured the level of Parkin phosphorylation by PINK1 in the presence of pUb using Phos-tag polyacrylamide gels. Under conditions where Parkin is minimally phosphorylated by PINK1, the addition of pUb robustly led to phosphorylation of roughly 50% of Parkin. The L266K Parkin

mutant, where the Ubl domain is released from RING1, was also phosphorylated even in the absence of pUb (Fig 3C). Moreover, the addition of pUb to L266K Parkin did not lead to an increase in phosphorylation. The phosphate group of pUb was required for the stimulation of Parkin phosphorylation. Addition of a non-phosphorylated ubiquitin (mutated to prevent its phosphorylation) did not increase the levels of Parkin phosphorylation (Fig EV3B). These results demonstrate that pUb binding to Parkin facilitates its phosphorylation by PINK1 due to release of the Ubl domain. Through competition for binding RING1, pUb binding, and Ubl phosphorylation reciprocally and synergistically lead to Parkin activation.

**pUb and Ubl bind RING1 at distinct sites**

The observation that the L266K mutation abrogates Ubl binding to R0RBR (Fig 2) but not pUb binding (Fig 3) suggests that the binding

sites for pUb and Ubl are distinct. To identify the pUb-binding site on Parkin, we mutated a series of residues in R0RBR and performed a pull-down assay with these mutants. We hypothesized the negatively charged phosphate on Ser65 would bind to a positively charged patch on R0RBR, and thus, we selected basic residues that were either located adjacent to the Ubl domain (Arg271, Arg275, Arg314, Arg334, K369), or bound sulfates in the Δ86–130 structure (K161, K211, H302, R305). Only mutation of His302 and Arg305 compromised binding to pUb (Figs 4A and EV4A), suggesting these two residues form the binding site for pUb. In the Δ86–130 structure, His302 and Arg305 are adjacent to the sulfate group with the lowest B-factor (Fig 1C). ITC measurements confirmed that mutation of His302 to alanine dramatically reduced the affinity of R0RBR

for pUb by 60-fold (22–1,333 nM), whereas mutation Lys211 in the other sulfate-binding site only reduced affinity by twofold (Figs 4B and EV4B). Together, these results confirm that the phosphate group of pUb binds to a patch on RING1 formed by His302 and Arg305.

We next conducted SAXS experiments to model the structure of R0RBR bound to pUb. Mass estimations from the Porod and $Q_r$ analyses were consistent with the molecular weights of both R0RBR (37 kDa) and the R0RBR–pUb complex (46 kDa) (Fig 4C). The $R_g$ values for both data sets were similar (Fig 4C), and the pair-distance distribution functions $P(r)$ have similar shapes (Fig EV4C), suggesting the R0RBR–pUb complex retained a compact structure. We then modeled the SAXS data using two alternative approaches. First, we

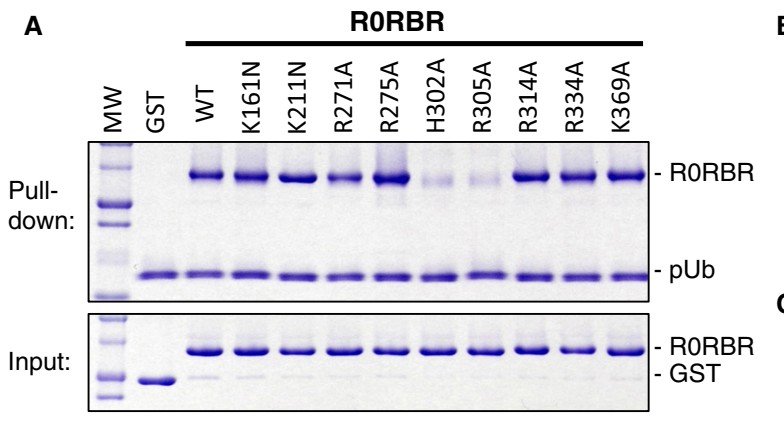

**B**

| Parkin construct | Affinity for pUb (nM) |
|---|---|
| H302A R0RBR | 1333 ± 90 |
| K211N R0RBR | 40 ± 15 |

**C**

| Parkin construct | Rg | Est. Mass (kDa) | |
|---|---|---|---|
| | | Porod | Qr |
| R0RBR | 27.3 ± 0.2 | 37.4 | 33.1 ± 0.4 |
| R0RBR + pUb | 27.3 ± 0.2 | 43.4 | 40.3 ± 0.3 |

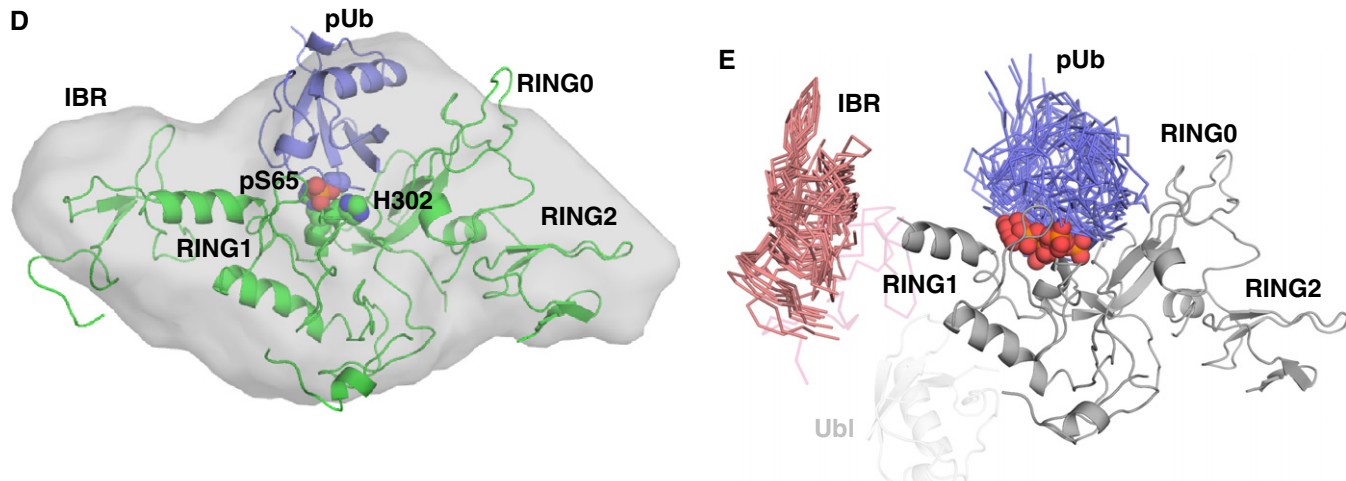

**Figure 4. pUb binds a RING1 site formed by His302 and Arg305.**

A  Pull down of Parkin R0RBR mutants using His-tagged pUb, immobilized on Ni-NTA agarose. Bound proteins were eluted, resolved by SDS–PAGE, and stained with Coomassie Blue.

B  ITC-derived affinities of pUb binding to two different Parkin R0RBR mutants representing the two major sulfate-binding sites.

C  SAXS-derived $R_g$ and mass parameters for R0RBR and the R0RBR–pUb complex.

D  *Ab initio* shape determination for the R0RBR–pUb complex (light gray surface), superposed with the R0RBR structure (green) and pUb (blue) docked to the His302 phospho-binding site.

E  Overlay of the ten best R0RBR–pUb SAXS models where phospho-Ser65 (red spheres) is bound to His302 and Arg305. The flexible IBR and pUb domains are shown as Cα traces colored in coral and blue, respectively. The original position of the IBR in the R0RBR structure is shown in light pink. For reference, the position of the Ubl domain in the Δ86–130 structure is shown in light gray.

performed *ab initio* shape determination to independently determine the location of the pUb-binding site. The shape determined from SAXS curves for the R0RBR module alone fit well to the X-ray crystal structure, thus validating the approach (Fig EV4D). The shape calculated from SAXS data of R0RBR–pUb revealed extra density located between RING1 and RING0 (Fig EV4E). The extra density could be accounted for by the placement of pUb with the phosphate in the sulfate-binding site next to His302 (Fig 4D). To confirm this model, we performed rigid-body modeling against the R0RBR:pUb SAXS data using the structures of the Parkin R0RBR module (Trempe *et al*, 2013) and pUb (Wauer *et al*, 2015b), while maintaining a short distance between Ser65 in pUb and His302/Arg305. This did not produce a docking model that satisfactorily fit the data ($\chi^2$ above 10), so we introduced an additional degree of freedom by allowing the IBR domain to move within a distance set by the length of the disordered linkers located before (G328-G329) and after (M380-Y391) the IBR domain. The rationale stems from the intrinsic flexibility of the IBR domain observed in previous crystal structures (Fig EV1E). The docking calculations with IBR movement converged satisfactorily with an ensemble of models very similar to the *ab initio* model (Fig 4E) and $\chi^2$ of $1.5 \pm 0.2$ (Fig EV4F and G). Identical docking calculations with pUb docked in the vicinity of Lys211 (the alternative sulfate-binding site) did not produce good fits.

These calculations suggest that pUb binds to the His302/Arg305 site, making contacts primarily with RING1 and RING0 (Fig 4D and E). The contacts on RING1 are opposite the Ubl-binding site and involve the helix that connects RING1 to the IBR domain (residues 309–327). That helix also contacts the Ubl domain (Fig 1F) and may mediate the allosteric coupling between pUb binding and Ubl release. The modeled pUb-binding site is also adjacent to RING1 residues Asp280-Gly284, which immediately follow the Ubl-binding helix (residues 261–274) (Fig 1E).

### Phosphorylation of Parkin stimulates its E3 ligase activity by increasing its affinity for the E2 enzyme

As an E3 ubiquitin ligase, Parkin needs to interact with an ubiquitin-charged E2 enzyme to conduct the ubiquitination of its substrates. Superimposition of the RING-UbcH7 structure (cCbl RING–UbcH7, pdb: 1FBV) onto the Δ86–130 Parkin structure predicts an overlap of both the Ubl domain and the REP helix with UbcH7 (Fig 5A). Therefore, we conducted ITC experiments to test how the phosphorylation of Parkin Ubl domain affects UbcH7 binding to Parkin. No binding and very weak binding ($> 700 \mu M$) were measured for full-length Parkin and the R0RBR module, respectively (Figs 5B and EV5A), consistent with our previous findings (Trempe *et al*, 2013). As expected, mutation of Trp403 in R0RBR to destabilize the REP:RING1 interaction enhanced binding of UbcH7 ($K_d = 119 \mu M$). Consistent with its reported increased activity, pParkin showed an increased affinity toward UbcH7 ($161 \mu M$) comparable to the R0RBR-W403A mutant (Figs 5B and EV5A). This suggests that phosphorylation of Ubl facilitates binding of UbcH7. We also tested the binding of pUb–pParkin complex with UbcH7 and obtained an even higher affinity ($31 \mu M$) than for pParkin alone, consistent with both phosphorylation events being required for full E3 ligase activity (Kane *et al*, 2014; Kazlauskaite *et al*, 2014b; Koyano *et al*, 2014).

NMR experiments were performed to validate our ITC results. Binding of Parkin to the E2 enzyme was monitored by the loss of signal in the NMR spectra of $^{15}$N-UbcH7 (Figs 5C–E and EV5B and C). Consistent with the previous results, no signal loss was observed for full-length Parkin. The R0RBR (Ubl deletion) or W403A mutants both showed increased binding to UbcH7 and their effects were additive, with the W403A R0RBR mutant displaying a greater signal loss than either single mutation. Mutation of T240R in RING1 (A240R in rat Parkin) prevented UbcH7 binding even in the context of activating mutations, thus confirming the model of E2 binding to RING1.

Titration with pParkin led to significant NMR signal loss comparable to the W403A R0RBR double mutant. Binding of pParkin or Parkin to UbcH7 could be further improved by the addition of pUb. However, the sole addition of pUb to R0RBR did not significantly affect binding to UbcH7, suggesting that the activating role of pUb is Ubl dependent. Phosphorylation of the Ubl domain had a much larger effect than simply removal of Ubl and following linker (Fig 5D and E).

To investigate the effect of Ubl and ubiquitin phosphorylation on the E3 ubiquitin ligase activity of Parkin, we performed autoubiquitination assays with unmodified or pre-phosphorylated Parkin in the presence of pUb. Addition of pUb mildly increased the low basal activity of wild-type Parkin, as observed by the formation of monoubiquitinated Parkin (Fig 5D). In contrast, phosphorylation of the Ubl (pParkin) dramatically increased its activity, which was not further enhanced by pUb. These results are consistent with our affinity measurements for UbcH7, which show that phosphorylation of Parkin leads to more significant E2 binding than pUb addition alone (Fig 5B–E). The N273K mutant, which cannot bind Ubl, behaves in a manner similar to wild type in the unmodified form and is even more active in the phosphorylated form. The implication is that Ubl phosphorylation plays a positive role in the stimulation of Parkin E3 ligase activity. Consistent with our pUb:Parkin binding model, we find that the pUb-binding site mutants H302A and R305A showed wild-type activity when phosphorylated, but the unmodified H302A mutant could not be stimulated by pUb. This confirmed that the H302A mutation specifically abrogated the ability of Parkin to bind pUb, but not its intrinsic E3 ligase activity.

## Discussion

Previous studies have shown that Parkin is natively inhibited through interdomain interactions and stimulated through two parallel activation steps: phosphorylation of Ser65 in the Parkin Ubl domain and pUb binding. Here, we show that the steps involve a switch between pUb binding and Ubl release (Fig 6). Despite their structural and sequence similarity, the Ubl domain of Parkin and ubiquitin is oppositely regulated by phosphorylation. Phosphorylation of the Ubl domain decreases its affinity for its binding site on RING1, while phosphorylation of ubiquitin increases its affinity.

Both activation steps involve conformational changes. The first is within the Ubl domain. The structure of Δ86–130 Parkin shows a large interaction surface between the β-sheet surface of the Ubl domain and the first α-helix in RING1 (Fig 1). Phosphorylation of Ser65 induces NMR chemical shift changes in the two central strands of the β-sheet surface and Asn8, which is more than 15 Å

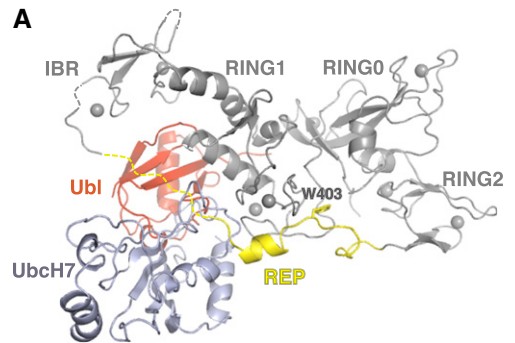

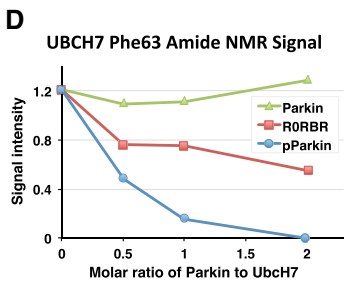

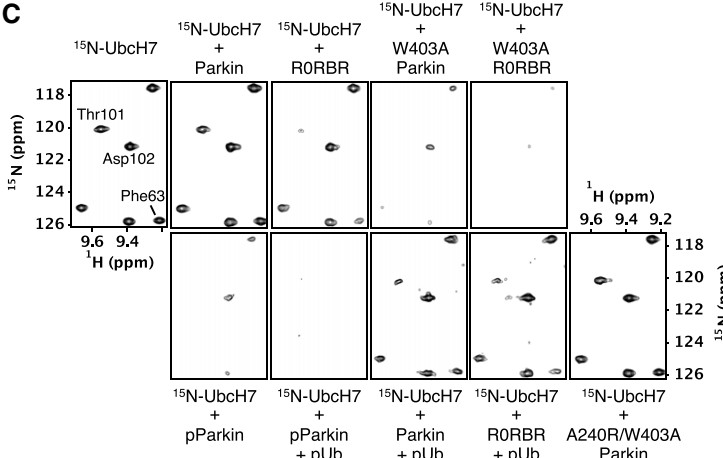

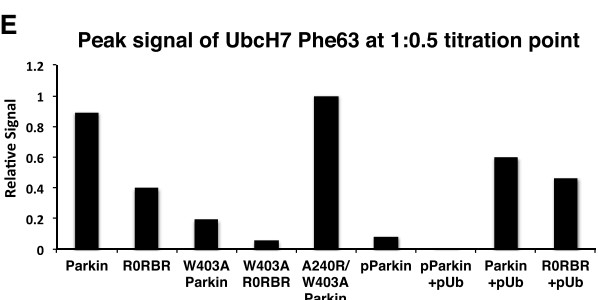

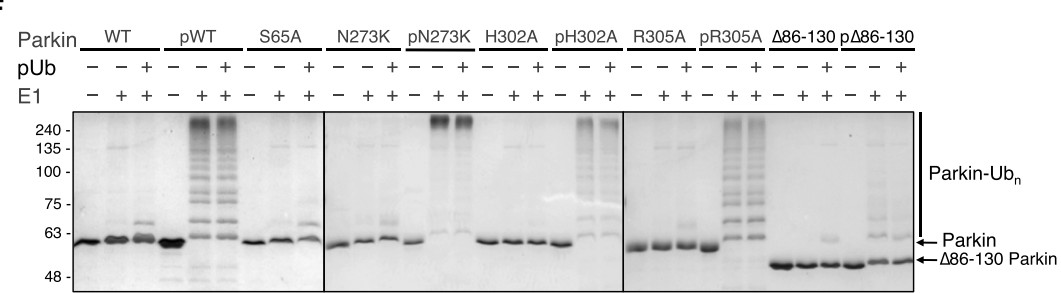

**Figure 5. Phosphorylation of Parkin and pUb binding increase the affinity of Parkin for UbcH7 and Parkin activity.**

A　UbcH7 (blue) modeled onto Parkin Δ86–130 (gray) based on the cCbl RING–UbcH7 complex. The Ubl and REP are colored in red and yellow, respectively.

B　ITC-derived affinities of different Parkin constructs titrated with UbcH7.

C　NMR analysis of Parkin binding to $^{15}$N-labeled UbcH7. $^{1}$H-$^{15}$N correlation spectra of UbcH7 (120 μM) were acquired in the presence of 120 μM of different Parkin constructs ± pUb. Peaks that belong to the RING1-binding surface of UbcH7 undergo the most broadening (signal loss) and are labeled in the first panel.

D　Signal intensity of $^{15}$N-UbcH7 the Phe63 backbone amide as a function of Parkin added. UbcH7 and Parkin concentrations were 1:0 (200 μM), 1:0.5 (120 μM:60 μM), 1:1 (120 μM:120 μM), 1:2 (90 μM:180 μM). Spectra were normalized for dilution of UbcH7 and the number of scans.

E　Relative signal intensity of $^{15}$N-UbcH7 Phe63 amide in the presence of different Parkin constructs and pUb complexes. UbcH7 concentration was 150 μM and the Parkin constructs and pUb concentrations were 75 μM. Peak heights were plotted relative to the UbcH7 spectrum with the E2 binding-deficient Parkin mutant A240R/W403A.

F　Parkin autoubiquitination assay. Unmodified or phosphorylated Parkin was incubated with ATP/Mg$^{2+}$ and UbcH7, with and without E1 enzyme and pUb. Products were resolved by SDS–PAGE and stained with Coomassie Blue. Ligase activity can be monitored by the loss of unmodified Parkin and the formation of higher molecular weight polyubiquitinated forms of Parkin.

from the site of phosphorylation (Fig 2E). These changes destabilize the Ubl–RING1 interface by 20-fold. Although phosphorylation of ubiquitin was found to induce the formation of a second minor form (Wauer *et al*, 2015b), this does not appear to be the case for the Ubl domain. Analysis of NOESY spectra did not find evidence

for a strand shift within the Ubl β-sheet. Presciently, molecular dynamics simulations of Parkin observed that phosphorylation of the Ubl domain led to its dissociation from RING1 as a result of changes in hydration and local structural changes (Caulfield *et al*, 2014).

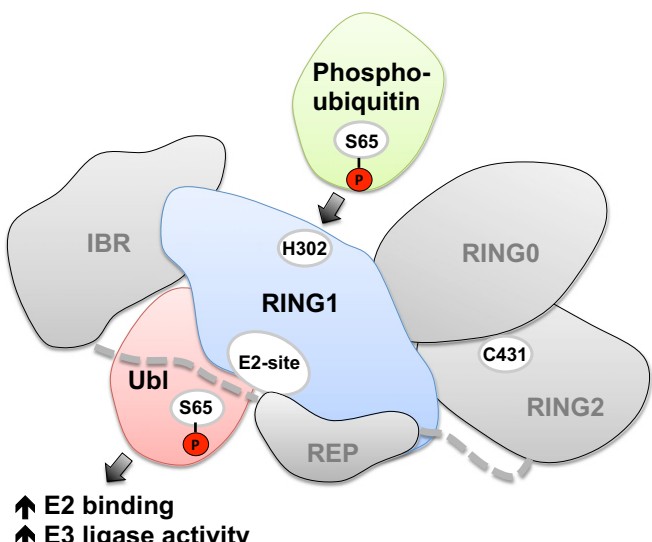

**E2 binding**
**E3 ligase activity**

**Figure 6.   Schematic of the Ubl/ubiquitin switch in the activation of Parkin.**
Phosphorylated ubiquitin binds to the H302 site on RING1 of Parkin and promotes the dissociation of the Parkin Ubl domain. The Ubl domain is in turn phosphorylated by PINK1 and activates Parkin.

The second conformational change is mediated through the Parkin R0RBR domain and gives rise to the antagonism between Ubl and pUb binding. As first observed by Ordureau *et al* (2014), Ubl phosphorylation increases the affinity of Parkin for pUb. Contrary to expectations, the two phosphoproteins do not compete for binding; the non-phosphorylated form of one competes with the phosphorylated form of the other. Two pieces of evidence suggest that this competition is due to a conformational change rather than overlapping binding sites. The first is that mutations that disrupt the Ubl–RING1 association do not prevent pUb binding. The N273K and L266K mutations block Ubl binding and actually increase pUb binding by displacing the Ubl domain. In the context of the R0RBR fragment, the L266K mutation has little effect on pUb binding (Fig 3B). The second is that mutations of residues over 15 Å away from the Ubl–RING1 interaction site prevent pUb binding. Disruption of a sulfate ion-binding site on the opposite face of the RING1 domain decreases pUb binding by at least 30-fold (Fig 4B). Two other groups have recently reported the identification of the same pUb-binding site (Kazlauskaite *et al*, 2015; Wauer *et al*, 2015a).

We used SAXS to gain insight into the conformational change that couples Ubl and pUb binding. In the model structures, the pUb sits on the RING1 domain opposite the Ubl-binding site on RING1. Alignment of the multiple structures of Parkin shows the IBR domain is flexible and varies in its position relative to the rest of Parkin (Fig EV1E). We were able to obtain excellent agreement between observed and theoretical SAXS curves of the complex of pUb and R0RBR but only when the IBR domain was allowed to move (Figs 4E and EV4G). The IBR–RING1 interface is the site of several Parkinson's disease missense mutations (R275W, Q311H, G328E), which may reflect a role in coupling pUb binding and Ubl release.

The catalysis by Parkin requires a third and larger conformational change to bring the incoming E2-conjugated ubiquitin molecule in contact with the Parkin catalytic residue, Cys431.

Native Parkin has very low affinity for E2 binding and low ubiquitin ligase activity. We previously showed that mutagenesis of the REP or deletion of the Ubl domain markedly increased UbcH7 binding and autoubiquitination activity (Trempe *et al*, 2013). Here, we observe that the physiological signal of Ubl phosphorylation has a stronger signal than either mutation alone. Parkin phosphorylation dramatically increases both UbcH7 binding and Parkin autoubiquitination activity (Fig 5). The response from pUb binding was more muted in both assays. The concomitant increase in E2-binding and ligase activity under a wide variety of conditions suggests that the binding of ubiquitin-charged E2 is the rate-limiting step in catalysis. Mutations and stimuli (phosphorylation) that increase E2 binding invariably increase autoubiquitination activity in a parallel fashion.

Both pUb binding and Ubl phosphorylation have been suggested to be the key, initiating step in Parkin activation on mitochondria (Kane *et al*, 2014; Kazlauskaite *et al*, 2014b; Koyano *et al*, 2014; Ordureau *et al*, 2014; Okatsu *et al*, 2015). Our switch model is evidence for feed-forward regulation in which the signals synergize with each other to promote activation. Binding of pUb releases the Ubl domain and promotes its phosphorylation by PINK1, and conversely, Parkin phosphorylation on the Ubl domain increases Parkin's affinity for pUb (Fig 6). However, the Ubl domain must do more than simply reduce pUb and E2 binding. Addition of isolated pUbl was shown to increase the activity of Parkin (Kazlauskaite *et al*, 2014b), and Ubl phosphorylation leads to exposure of the active site cysteine while the addition of pUb did not (Ordureau *et al*, 2014). In cells, ΔUbl Parkin does not recruit efficiently to mitochondria and, in patients, missense mutations that disrupt the Ubl domain lead to Parkinson's disease. We observed that phosphorylation of the Ubl domain had a much stronger effect on E2 binding than its removal or addition of pUb (Fig 5). pUbl likely plays a role in stabilizing the Parkin–E2 complex and the conformational change that allows the transfer of ubiquitin onto Cys431.

Thus, a model of Parkin activation in mitophagy is emerging where pUb binding plays a lead role in recruiting Parkin to mitochondria and pUbl plays a larger role in regulating Parkin activity. Following the initial phosphorylation by PINK1 of ubiquitin on mitochondrial proteins, Parkin is recruited and the Ubl domain released through the switch-like mechanism described here. Phosphorylation of the Ubl by PINK1 locks Parkin to the mitochondria surface by increasing the affinity for pUb and derepresses Parkin ubiquitin ligase activity. The resulting ubiquitination of mitochondrial proteins leads to additional rounds of Parkin recruitment and engagement of the autophagic machinery. More studies are needed to clarify the extent to which Parkin is required for the initial ubiquitination of mitochondria and the residual activity of non-phosphorylatable S65A Parkin in cellular recruitment assays (Kane *et al*, 2014; Ordureau *et al*, 2015).

## Materials and Methods

### Cloning, expression, and purification of recombinant proteins

PCR mutagenesis was used to generate Parkin single-point mutants. Protein expression and purification of the different Parkin variants and UbcH7 were done in BL21 (DE3) *E. coli* cells using conditions previously described (Trempe *et al*, 2013). Briefly, proteins were

purified by glutathione–Sepharose or Ni-NTA agarose affinity, followed by 3C cleavage and size-exclusion chromatography in various buffers used in various techniques. GST-TcPINK1 (143–570) was purified in a similar manner to Parkin, as previously described (Koyano *et al*, 2014). [15]N-labeled UbcH7 was produced in M9 minimal medium supplemented with [15]NH$_4$Cl. Phosphorylated ubiquitin was produced and purified as described (Wauer *et al*, 2015b) using GST-TcPINK1. Phosphorylated Ubl and Parkin were produced and purified according to a published procedure (Ordureau *et al*, 2014). Protein concentrations were determined using UV and amino acid analysis.

## Crystallization

Crystals of Δ86–130 Parkin were grown at 4°C using hanging drop vapor diffusion method by mixing 1 μl of protein at 11 mg/ml in 15 mM Tris–HCl pH 8.0, 10 mM DTT, and 1 μl of mother liquor containing 0.3 M ammonium sulfate, 0.1 M HEPES pH 7.5, 18% (w/v) PEG 3350, 56 mM β-mercaptoethanol. Crystals appeared and reached their maximal size of 50 × 50 × 100 μm after 2 days. Crystals were cryoprotected in a solution of MiTeGen LV CryoOil before being flash-frozen in liquid nitrogen.

## Data collection and structure determination

Diffraction data for Δ86–130 Parkin were collected in house on Rigaku MicroMax-007 HF rotating anode source equipped with Rigaku VariMax HF monochromator. A total of 644 images with an oscillation angle of 0.5° were measured at the Cu K-edge on a Rigaku Saturn 944+. Reflections were integrated using XDS (Kabsch, 2010b) and scaled with *XSCALE* (Kabsch, 2010a). The high-resolution cutoff was assessed using the CC* correlation coefficient as described (Karplus & Diederichs, 2012). The crystal exhibited merohedral twinning in the H3 space group, approximating a H32 space group. The structure was solved by molecular replacement using Molrep (Vagin & Teplyakov, 2010) as part of CCP4 suite (Winn *et al*, 2011), using atomic coordinates of the R0RBR (PDB 4K7D) and the Ubl (PDB 2ZEQ) as search models. The asymmetric unit in the H3 space group contained two molecules. Packing analysis shows a layer arrangement along the *z*-axis of H3 lattice. Interactions within each layer are very extensive, but interactions between layers are made of the Ubl domains only, which do not interact with each other and fill the gaps along the primitive H3 axis direction. Electron density was of good quality at this stage, and model building was performed using the program *COOT* (Emsley *et al*, 2010). The structure was refined using *REFMAC5* (Murshudov *et al*, 1997), using TLS refinement. Coordinates and structure factors of the Δ86–130 Parkin crystal structure were deposited in the Protein Data Bank under accession code 4ZYN (Table 1).

## Small-angle X-ray scattering (SAXS)

Small-angle X-ray scattering data were collected on SIBYLS beamline (12.3.1) at the Advanced Light Source at Lawrence Berkeley National Laboratory. Scattering data were collected for 0.5–1 s at 20°C at protein concentrations of 2, 4, and 8 mg/ml for Δ86–130 Parkin and L266K Parkin and at 1.5, 3, and 6 mg/ml for R0RBR, R0RBR–pUb complex, full-length Parkin, pParkin, pParkin–pUb complex, and W403A Parkin. Background scattering from the buffer (50 mM

Tris–HCl pH 8.0, 150 mM NaCl, 2% glycerol, 10 mM DTT) was measured for 0.5–1 s. Scaling, buffer subtraction, and merging were performed using ATSAS (Petoukhov *et al*, 2012). The merged scattering curve was fitted to individual chains in the crystal structure using *CRYSOL* (Svergun *et al*, 1995). The pair-distance distribution were calculated using *GNOM* (Svergun, 1992). Molecular mass was estimated using the Porod volume method (Petoukhov *et al*, 2012) or the Q$_R$ method (Rambo & Tainer, 2013).

*Ab initio* shape calculations were performed using the program *GASBOR* against intensity in reciprocal space (Petoukhov *et al*, 2012). Forty models were generated with $\chi^2 < 2$ and averaged using *DAMAVER* (Petoukhov *et al*, 2012). The R0RBR structure (pdb 4K7D, chain A) was superposed to the averaged *GASBOR* model using *SUPCOMB* (Petoukhov *et al*, 2012). Rigid-body modeling of the R0RBR–pUb complex was performed with *CORAL* (Petoukhov *et al*, 2012), using the crystal structure of R0RBR (PDB 4K7D, chain A) and the major conformation of pUb (PDB 4WZP, chain A) as input models. Calculations were performed with two different distance restraints based on the conserved sulfate-binding sites: Ser65 was set to be within 10 Å (Cα-Cα) of either H302/R305 or K161/K211. Because the SAXS data could not be fitted by simple rigid-body fits, the IBR domain was allowed to move. Flexible linker residues were defined as G328-G329 (between RING1 and IBR), and M380-Y391 (between IBR and REP). Forty models were generated, and the 10 models with the lowest $\chi^2$ (< 2) and penalty (< 1) were selected.

## Isothermal calorimetric assay

ITC measurements were carried out at a constant temperature of 20°C using ITC200 (Malvern). Samples were in 50 mM Tris–HCl pH 7.4, 150 mM NaCl, 1 mM TCEP. Data were analyzed using Origin v7 software. Protein concentrations in the cell and syringe are indicated in the figures. The stoichiometry was determined experimentally for Ubl and pUb titrations (Figs 2A and 3A) and generally refined to values between 0.8 and 1.0. The stoichiometry was fixed to 1:1 for UbcH7 titrations (Fig EV5A).

## Pull-down assays

*In vitro* pUb pull-down assays with R0RBR variants (Fig 2A) were performed using 1 nmol of His-ubiquitin previously phosphorylated by PINK1 and 25 μl of Ni-NTA resin (Qiagen). The His-bound resin was incubated for 15 min with 1 nmol of R0RBR wild type, K161N, K211N, R271A, R275A, H302A, R305A, R314A, R334A, K369A, and GST, in TBS with 2 mM BME with 0.02% [v/v] Igepal 630. The resin was washed with 3 × 1 ml of buffer for 30 s and eluted with 25 μl of SDS–PAGE loading buffer containing 0.3 M imidazole pH 8.0. The products were resolved using 12% Tris-tricine SDS–PAGE and stained with Coomassie Brilliant Blue.

*In vitro* Ubl/pUbl pull-down assays (Fig EV2B) were performed using 2 nmol of GST-R0RBR/GST and 15 μl of glutathione–Sepharose resin (Pierce). The GST-bound resin was incubated for 15 min with 2.5 nmol of Ubl or pUbl in TBS with 2 mM DTT and 0.02% [v/v] Igepal 630. The resin was washed twice with 1 ml of buffer for 30 s and eluted with 15 μl of SDS–PAGE loading buffer. The products were resolved using 15% Tris-glycine SDS–PAGE and 14% Tris-tricine SDS–PAGE (20 μM Phostag, 40 μM ZnCl$_2$) and visualized with Coomassie stain.

## Kinase assay

Parkin phosphorylation assays were performed with 0.1 μM GST-TcPINK1 (143–570), 50 mM Tris–HCl, 100 mM NaCl, 1 mM DTT, 2 mM MgSO$_4$, and 1 mM ATP in a reaction volume of 25 μl at 30°C. The assays in Fig 3C included 3 μM Parkin wild type, L266K, or H302A as substrates in the presence or absence of 3 μM pUb. The assays in Fig EV3B included 30 μM Parkin wild type in the presence or absence of 30 μM His-tagged pUb or 30 μM His-tagged UbS65A. Reactions were quenched by the addition of SDS–PAGE loading buffer. Aliquots of 20 μl (Fig 3C) or 2 μl (Fig EV3B) of the reactions were loaded on 10% Tris-glycine gels containing 20 μM Phos-tag and 40 μM MnCl$_2$ and visualized with Coomassie stain.

## NMR spectroscopy

$^{15}$N-labeled UbcH7, Ubl, pUbl, and pUb and unlabeled Parkin constructs (full-length, R0RBR, W403A Parkin, L266K R0RBR, W403A R0RBR, pParkin, and A240R/W403A Parkin) were buffer-exchanged into NMR buffer (20 mM Tris–HCl, 120 mM NaCl, 2 mM DTT, pH 7.4). All samples were processed in a similar manner, and the pH was the same for all samples used in comparisons. pUb was added to pParkin and Parkin in excess (1:2 molar ratio) to form the complex. The pUb–R0RBR complex was purified by size-exclusion chromatography in NMR buffer. Titrations were performed by adding unlabeled Parkin proteins to 0.03–0.2 mM $^{15}$N-labeled proteins to obtain the molar ratios indicated. $^{1}$H-$^{15}$N correlation spectra were acquired at 293 K at field strengths of 600 MHz (Figs 5D and EV5C) and 800 MHz (Figs 2, 3, 5C and E, and EV5B). Spectra were processed using NMRpipe (Delaglio *et al*, 1995) and analyzed with NMRView J (One Moon Scientific) and adjusted to account for differences in concentration and acquisition time. $^{15}$N-$^{1}$H backbone assignments for UbcH7 were obtained from the BMRB entry 15498 (Serniwka & Shaw, 2008). $^{15}$N-$^{1}$H backbone assignments for Ubl and pUbl were obtained using $^{15}$N-NOESY/TOCSY experiments for $^{15}$N-labeled proteins and HNCACB and CBCA(CO)NH for $^{13}$C,$^{15}$N-pUbl. Weighted chemical shift perturbations were calculated according to the equation ($\Delta HN^2 + (0.2*\Delta N)^2)^{1/2}$.

## Ubiquitination assay

Ubiquitination assays were performed for 45 min at 37°C in the presence of 50 mM Tris–HCl pH 7.5, 120 mM NaCl, 1 mM DTT, 4 mM ATP, 50 μM ubiquitin, 10 mM MgCl$_2$, 50 nM E1, 2 μM UbcH7, and 2.5 μM Parkin with or without 5 μM of pUb. Reactions were stopped with the addition of SDS–PAGE sample buffer containing 100 mM DTT and analyzed by gel electrophoresis and Coomassie staining.

**Expanded View** for this article is available online:
http://emboj.embopress.org

## Acknowledgements

We thank Cheryl Arrowsmith for the His-UbcH7 construct, Richard Youle for the His-ubiquitin and His-S65D ubiquitin constructs, Dmitry Rodionov for help with in-house X-ray data collection, the Advanced Light Source for access and data collection on the SIBYLS high-throughput SAXS beamline, and group members for insights and helpful discussion. We acknowledge support from Parkinson Society Canada and the Canada Research Chairs Program (J-FT) and from Canadian Institutes of Health Research grant MOP-125924 (KG).

## Author contributions

VS performed construct design, protein purifications, SAXS data acquisition, ITC assays, pull-down assays, and wrote the manuscript. AL performed protein purifications, Δ86–130 Parkin crystallization and data collection, SAXS data acquisition, and ITC assays. MS and GK purified Parkin and performed NMR titrations. MV performed the ubiquitination assay. SR did the kinase assays. TS performed NMR titrations and analysis. JW did X-ray data processing and phasing. J-FT did crystal structure refinement, SAXS data analysis and with KG conceived experiments, performed data analysis, and wrote the manuscript.

## Conflict of interest

The authors declare that they have no conflict of interest.

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
