## [Review Process File · The EMBO Journal]

Manuscript EMBO-2015-92237

A Ubl/ubiquitin switch in the activation of Parkin

Veronique Sauve, Asparouh Lilov, Marjan Seirafi, Marta Vranas, Shafqat Rasool, Guennadi Kozlov, Tara Sprules, Jimin Wang, Jean-Francois Trempe and Kalle Gehring

Corresponding author: Kalle Gehring, McGill University

Review timeline:	Submission date:	07 June 2015
	Editorial Decision:	29 June 2015
	Revision received:	21 July 2015
	Accepted:	24 July 2015

Editor: Karin Dumstrei

Transaction Report:

1st Editorial Decision

29 June 2015

Thank you for submitting your manuscript to The EMBO Journal. Your study has now been seen by three referees and their comments are provided below.

As you can see the referees appreciate the structural analysis on Parkin and are supportive of publication here. They raise a number of different points that are clearly outlined below. Referee #2 would like to see more ubiquitination and ITC assays while referee #1 also requests further in vivo and functional assays. Referee #3 raises minor concerns only. I see the importance of adding more in vivo data and if you have such data please do include, but I will also not insist on the inclusion of such data for publication here. I would like to ask you to focus your revision efforts on the ubiquitination assays, ITC and mutational analysis.

I would be very helpful if you could let me know the timeframe of doing the revisions as I am keen on getting the revised version back as soon as possible.

When preparing your letter of response to the referees' comments, please bear in mind that this will form part of the Review Process File, and will therefore be available online to the community. For more details on our Transparent Editorial Process, please visit our website: http://emboj.embopress.org/about#Transparent_Process

You can use the link below to upload the revised version

REFeree REPORTS

Referee #1:

PINK1 activates Parkin E3 ligase activity through two phosphorylation events on S65 of the N-terminal ubiquitin-like (UBL) domain of Parkin and of ubiquitin itself. Currently, it is believed that UBL phosphorylation relieves Parkin autoinhibition, while phosphoubiquitin accelerates discharge of UbcH7~ubiquitin. The authors have previously reported a low resolution 6.5 Å structure of full-length Parkin, revealing how it adopts an autoinhibited conformation, in which RING0 masks C431 and the REP domain prevents E2 binding to RING1. Here, they report a higher resolution 2.5 Å crystal structure of full length Parkin lacking residues 86-130 in the flexible linker between the UBL and the rest of the protein, which provides further insights into the cooperation of these two phosphorylation events in Parkin activation. Importantly, a potential pS65 Ub-binding site and a UBL-RING1 interface were identified, involving H302/R305 and L266, respectively. Mechanistically, pUb binding promotes the dissociation of the UBL from the RING1 domain, while UBL phosphorylation decreases its affinity for RING1. pUb binding and UBL phosphorylation together enhance the binding of the UbcH7 E2 and increase Parkin E3 ligase activity.

Overall, this manuscript contains some new structural insights that enhance our understanding of the biochemical properties of Parkin that will be of interest to the ubiquitin community. However, there are a number of points that would need to be addressed before publication.

Major issues are:

1. The only mutations that define the putative pUb-binding pocket are H302A and R305. Analysis of additional mutations to characterize the proposed pUb-binding pocket more completely would be desirable, e.g. K151.
2. The biochemical data support the relevance of the identified pUb-binding pocket, but additional *in vivo* analysis is needed to establish the importance of the proposed pUb-binding pocket in Parkin's function in recognizing damaged mitochondria in cells. For instance, the authors need to express and test the efficiency with which H302A (or R305A) Parkin is recruited to damaged mitochondria in CCCP-treated cells compared to WT Parkin.
3. It remains unclear how pUb binding to this pocket "activates" RING1 to promote E2~Ub discharge/transfer to C431?
4. Is there an explanation for why the phosphorylated UBL domain itself does not bind to the pUb-binding pocket; with the long linker between the UBL domain and RING0 there seems no obvious reason why it should not do so. In this regard, the Komander group has recently reported that phosphorylation of S65 in Ub can generate an alternative conformation in solution (Wauer and Komander, EMBO J 32:2099). Is there any evidence that the same is true of pS65 UBL, and could this explain why it does not bind to the pUb-binding pocket?

Points: 1. The authors need to demonstrate the functionality of Parkin Δ 86-130 compared to full length Parkin, both by biochemical ubiquitylation assays *in vitro*, and by functional assays in a mitophagy model. They could also determine whether Parkin Δ 86-130 C431S is still able to trap ubiquitin and be activated by pUb in a ubiquitylation assay to the same extent as WT Parkin.

2. Figure 3C: Although pUb or the L266K mutation was able to increase Parkin S65 phosphorylation by PINK1, the difference is not significant, suggesting that the release of the UBL domain is not a prerequisite for S65 phosphorylation. The PINK1 kinase assay was a 30 min reaction, which may be too long, and therefore mask the importance of pUb binding in priming UBL phosphorylation. The authors should try a shorter time point, and use a pS65-specific antibody to quantify phospho-Parkin.

3. To identify the potential pUb binding site, the author hypothesized that the negatively-charged phosphate on S65 would bind to a positively charged patch on R0RBR, and thus they selected basic residues that were either located adjacent to the UBL domain (R271, R275, R314, R334, K369), or were bound to sulfates in the Δ 86-130 structure (K161, K211, H302, R305). Do any other positively charged patches exist in regions outside of RING1+IBR? Mutation of H302 and R305 compromised pUb binding to R0RBR (Figure 4).

4. Figure 5C: These data show that addition of pUb to R0RBR did not affect UbcH7 binding. This is an interesting observation, as R0RBR can bind to pUb with high affinity (Figure 3A). As the authors suggested, this experiment indicates that UBL domain has a positive role in Parkin E3 activity. The affinity of R0RBR+pUb and R0RBR+pUb+pUBL on UbcH7 binding (and Ub discharge) should be measured and included in Figure 5B. This point may be a significant contribution of this paper, since there are hints in the literature that UBL domain may not simply play a negative role in Parkin activity. For instance, the UBL domain has been shown to be essential for ubiquitin trapping in Parkin C431S (Zheng and Hunter, Cell Res 23:886). Moreover, the pUBL domain can stimulate the E3 activity of UBL-deleted Parkin (Kazlauskaite et al. Biochem J 460:127) and is essential for Miro ubiquitylation (Kazlauskaite et al. Open Biol 2:120080).

5. Figure 5D: In the pWT panel, it appears that addition of pUb did not further enhance the E3 activity of pParkin, apparently arguing against the importance of pUb. Why did pWT+pUb have lower E3 activity compared to WT+pUb, considering that WT Parkin has a low affinity for pUb and S65D ubiquitin did poorly in WT Parkin binding (Fig. EV3)? To the reviewer, the results seem to support the idea that the H302A mutation makes the UBL phosphorylation essential for Parkin activity, with the two phosphorylation events being redundant rather than additive and cooperative. Here, there are some issues that the authors may take into consideration. First, does the Parkin autoubiquitylation activity really reflect the E3 activity toward its substrates? This could easily be tested. Second, does pUb inhibit Ub chain formation? In vivo, Parkin increases K63 chain formation on the damaged mitochondria, and because pS65 lies very close to K63, it may therefore affect the chain formation by Parkin. A C-terminal GG mutant form of pUb might be a better option. Moreover, a C431S mutant should be used as negative control to confirm that the autoubiquitylation activity occurs through a HECT-like mechanism.

Minor points: 1. If equimolar pUb can compete the binding of UBL domain to R0RBR (Figure 3B), why did S65D ubiquitin not pull down WT Parkin (Figure EV3A)?

2. Page 6: Fig. EV1D and Fig. EV1E are mislabeled; there is no EV1E.

Referee #2:

Parkin is a key player in the mitochondrial quality control pathway and mutations in Parkin cause early-onset Parkinsons disease. Parkin has been shown to exist in an autoinhibited state that needs to be released to promote ligase activity and induce mitophagy. Release of autoinhibition is promoted by phosphorylation events induced by PINK1 that alter the overall structure of Parkin to allow interaction with its cognate E2 and ubiquitination of substrate proteins.

The manuscript by Sauve and colleagues describes the crystal structure of full-length Parkin from which a loop region has been deleted (delta 86-130) and provides an extensive biophysical characterization of Parkin and its interaction with phosphorylated ubiquitin which allow the authors to propose a location for the pUb binding site on Parkin. Furthermore, they explore the role of Parkin phosphorylation and pUb binding on the interaction with UbcH7 and based on their data present a model for Parkin activation by PINK1.

This is an interesting study that makes an important contribution to advancing our understanding of Parkin activation through phosphorylation of its Ubl domain and binding of pUb. It provides a first insight into the conformational changes occurring during the activation process which are likely to be highly dynamic and hence difficult to capture by crystallographic studies alone.

Major concerns

- The authors need to carry out ubiquitination assays to test if the crystallized fragment is indeed a good mimic of the wt protein or if the deleted loop may play an additional role that is not understood at the moment.

Generally, the paper would profit from carrying out ubiquitination assays with mutants of those residues that have been identified to play an important role (eg L266 or H302) to test if their activity supports the function that has been ascribed to them.

Similarly, it would be useful to link the ITC studies to ubiquitination assays to highlight how changes in affinities translate into changes in activity.

- Figure 2B: the comparison of spectra from the 1:1 and 1:2 pUbl + R0RBR mixtures clearly shows that there is a signal reduction in the 1:2 mixture, indicating that an interaction is taking place, though a weak one. Could the authors try to go to higher ratios and try to estimate a K_d for the interaction?

Why do the authors show different regions for the HSQC spectra of phosphorylated and unphosphorylated Ubl domains?

- The authors use the differences in D_{max} values of wt full length Parkin compared to pParkin and the L266K mutant as an indication that their conformation changes and that the molecules have become elongated due to dissociation of the Ubl domain.

However, in figure EV4E where the shape of R0RBR and its pUb complex are compared a similar difference in D_{max} is regarded as "only slightly different". It's either one or the other. Please explain and rephrase.

- The pull down assays shown in figure 3EV3A are not very convincing:

The ITC data show that pParkin binds to pUb with an affinity of 17nM but there's hardly any interaction seen in the pull down between S65E Parkin and S65D ubiquitin. Does this mean that the mutants are not very good mimetics of the phospho forms? There's a ~15 fold difference in affinities between S56E Parkin and pParkin for pUb, but the complex should be still tight enough to be pulled down together.

Why did the authors do the pull downs with the phosphomimetics whereas the ITC experiments were done with the phospho forms? I am also surprised that a complex that has an affinity of ~0.4 μ M (wt Parkin + pUb/S65D Ub) cannot be detected in a pull down assay. Overall, I fail to see what the pull down experiments add to the extensive characterization by ITC and they should be removed unless the authors think they add more information. If this is the case please discuss.

- I'm not convinced by the approach taken to derive the model presented in figure 4D and figure 4C isn't very meaningful either. It would be much more useful if the authors modelled a molecular envelope for the R0RBR/pUb complex based on their experimental data and then fitted the R0RBR structure into this envelope. This should indicate where the binding site for pUb is - ideally close to H302.

Please add the scattering data for R0RBR+Ub to figure EV4C.

Minor concerns

- A 2.5 Å crystal structure is not "high-resolution".

- page 6: Please add an explanatory sentence to the main text explaining how SAXS experiments exclude model 2 as the figure and associated text are only shown in supplementary material.

- How often have the ITC titrations been repeated and how have the error estimates given been calculated?

- page 6: Where is figure EV1E? Should this read EV1D?

- It would be helpful for the reader to highlight the E2-binding site in figure 1B.

- page 7: The authors should add a figure (could be supplementary) to illustrate the point made about the flexibility of the IBR domain and the different positions it adopts. Do the authors think this is physiologically relevant?

- What is the point the authors want to make with figure EV2A? These titrations have already been shown in figure 2.

And why did the authors feel they needed to confirm data from ITC and NMR experiments with a pull down assay, especially one that is not a very good one (figure 2EV2B)? This assay doesn't add anything and should be removed.

- bottom page 8: Please say explicitly how SAXS provides shape information through the calculation of the pair-distance distribution so that a non-expert can follow the argument.
- page 9: Please add a sentence explaining what is meant by "an alternative conformation of the Ubl domain". Please also say explicitly that the perturbations were mapped onto the Ubl domain of the delta68-130 Parkin structure.
- Figure 3B: I am slightly surprised that the addition of 1 equivalent R0RBR to a 100 μ M solution of Ubl leads to an almost complete loss of signal given that their affinity is only 22 μ M. Are the authors sure that the concentrations reported are correct?
- Please show all the ITC titrations that have been combined in a given figure on the same scale for easier comparison.
- A cartoon explaining the conformational changes and model described in the discussion would be very helpful.

Referee #3:

This manuscript integrates multiple approaches to provide fundamental information on how parkin activity is regulated by phosphorylation of the Ubl domain and of ubiquitin. A 2.54Å resolution crystal structure of full length parkin with a linker region deleted is solved and SAXS, NMR, ITC and site-directed mutagenesis used to characterize the effects of Ubl and ubiquitin phosphorylation. This reviewer is enthusiastic toward this manuscript, which is compelling with multiple elements that weave together in an elegant way. Some technical concerns however are listed below that should be addressed, although can probably be without further experiments.

1. The authors should explicitly state whether the NMR data are collected and processed in an identical manner when being compared to each other. For example, in Fig 2C, it's probably true that the sample concentration, experimental parameters (especially number of scans), and processing parameters are identical for the left and center panel, but this is not stated.
2. It seems that all NMR experiments are done at pH 7.4. This is not ideal for observing amide groups because of exchange with water. Moreover, is the pH reported for buffer or final sample? As stated above, the authors should explicitly state that the pH is not changed in comparisons in Materials and Methods if this is the case. Confidence would be higher if 1D traces were included showing broadened lineshapes rather than just disappearance of signals. A couple of traces could be shown as a figure insert even.
3. Regions of the 15N NOESY should be displayed to support the authors' conclusion of no changes in beta-strand alignment following Ubl phosphorylation.
4. The HSQC spectra corresponding to the chemical shift perturbation for pUbl vs Ubl should be included.
5. The authors need to address further what the changes to NMR signals from Ubl phosphorylation reflect. Clearly they do not think that these changes are from secondary structure changes based on the NOESY spectrum. The authors write however that they "are consistent with structural changes," thus potentially conveying a mixed message. Amides are highly sensitive to their chemical environment and can shift by changes in dynamics, reconfiguration of interactions with neighboring amino acids, etc, as the authors probably know but their readers may not. The authors should indicate what their data collectively best reflects.

Referee #1:

PINK1 activates Parkin E3 ligase activity through two phosphorylation events on S65 of the N-terminal ubiquitin-like (UBL) domain of Parkin and of ubiquitin itself. Currently, it is believed that UBL phosphorylation relieves Parkin autoinhibition, while phosphoubiquitin accelerates discharge of UbchH7~ubiquitin. The authors have previously reported a low resolution 6.5 Å structure of full-length Parkin, revealing how it adopts an autoinhibited conformation, in which RING0 masks C431 and the REP domain prevents E2 binding to RING1. Here, they report a higher resolution 2.5 Å crystal structure of full length Parkin lacking residues 86-130 in the flexible linker between the UBL and the rest of the protein, which provides further insights into the cooperation of these two phosphorylation events in Parkin activation. Importantly, a potential pS65 Ub-binding site and a UBL-RING1 interface were identified, involving H302/R305 and L266, respectively. Mechanistically, pUb binding promotes the dissociation of the UBL from the RING1 domain, while UBL phosphorylation decreases its affinity for RING1. pUb binding and UBL phosphorylation together enhance the binding of the UbchH7 E2 and increase Parkin E3 ligase activity.

Overall, this manuscript contains some new structural insights that enhance our understanding of the biochemical properties of Parkin that will be of interest to the ubiquitin community. However, there are a number of points that would need to be addressed before publication.

We would like to thank the reviewer for his encouraging comments. We have performed a number of the requested experiments to confirm and extend our conclusions. We believe that the revised manuscript is significantly improved and trust that the reviewers will find it suitable for publication.

Major issues are:

1. The only mutations that define the putative pUb-binding pocket are H302A and R305. Analysis of additional mutations to characterize the proposed pUb-binding pocket more completely would be desirable, e.g. K151.

As other groups have identified additional mutations that block pUb binding, we have focused on expand our analysis of ubiquitination and E2 binding assays. As described below, these experiments suggest a hierarchical ordering of the phosphorylation steps that lead to Parkin activation.

2. The biochemical data support the relevance of the identified pUb-binding pocket, but additional in vivo analysis is needed to establish the importance of the proposed pUb-binding pocket in Parkin's function in recognizing damaged mitochondria in cells. For instance, the authors need to express and test the efficiency with which H302A (or R305A) Parkin is recruited to damaged mitochondria in CCCP-treated cells compared to WT Parkin.

While the reviewer raises an important point, the requested cellular studies would greatly expand the scope of the current manuscript, which is currently focused on biochemical and structural characterization of parkin. Cellular recruitment assays could take several months to complete and are better suited for a follow-up study that could more completely address the relative importance of Ubl and Ub phosphorylation pathways. With a total 11 figure and extended view figures, we feel that the manuscript is better served by focusing on the biophysical experiments that show the antagonistic binding of pUbl and pUb to parkin. In the revised manuscript, we report improved autoubiquitination assays, new PINK1 kinase assays, and additional NMR experiments.

3. It remains unclear how pUb binding to this pocket "activates" RING1 to promote E2~Ub discharge/transfer to C431?

We agree that the structure of the catalytic intermediate involved in ubiquitin transfer is unknown. A large number of groups are actively pursuing that structure. While not addressing that question, our work is an important step forward that answers questions about the earlier events in the activation process: the binding of pUb and phosphorylation of the Ubl domain.

Our NMR data (Figure 5C) shows that the binding of pUb to RING1 only slightly improves E2 binding to RING1. The effect of pUb binding on E2 binding is indirect. The E2 binding site on RING1 is occluded by both the Ubl and the linker between the IBR and the REP (Figure 5A). Our data show that pUb binding to Parkin promotes the release of Ubl from RING1 (Figure 3B) and enhances its phosphorylation (Figure 3C). The phosphorylation of Ubl then weakens its interaction with RING1 and dramatically increases affinity towards UbcH7, which can also be achieved by deleting the Ubl and mutating the REP (Figure 5B-E). Moreover, our ubiquitination assays shows that it is, phosphorylation of Parkin, not pUb binding, that leads to the most dramatic increase in activity (Figure 5F). Moreover, pUb does not further stimulate activity of phosphorylated Parkin (Figure 5F). Phosphorylated Parkin binds better to E2 and thus promotes E2~Ub transfer, but how this is achieved at the structural level is unknown.

4. Is there an explanation for why the phosphorylated UBL domain itself does not bind to the pUb-binding pocket; with the long linker between the UBL domain and RING0 there seems no obvious reason why it should not do so. In this regard, the Komander group has recently reported that phosphorylation of S65 in Ub can generate an alternative conformation in solution (Wauer and Komander, EMBO J 32:2099). Is there any evidence that the same is true of pS65 UBL, and could this explain why it does not bind to the pUb-binding pocket?

The weaker binding of pUbl appears to be due small structural adjustments. NMR ¹⁵N-HSQC-NOESY experiments show no major differences in interstrand NOEs in the beta-sheet region of Ubl and pUbl (see new figure EV2G). This argues against pUbl adopting the minor conformation observed in pUb by Wauer and Komander. Previous molecular dynamics simulations of parkin have predicted weaker binding of pUbl to the Ubl-binding site as a result of changes in hydration and local structural changes (Caulfield et al, 2014).

There are a number of differences between Parkin Ubl and Ub that could explain why pUbl does not bind the pUb-binding site. Despite having a similar 3D structure, Ubl and Ub share only 30% sequence identity. Different charge and hydrophobic distribution on their surface due to their different sequence could dictate their differential affinity for the Ubl and pUb binding sites. For example, the conserved Asn8 in Parkin, which is involved in Ubl binding to RING1 (Figure 1D) and which is perturbed upon Ubl phosphorylation (see Figure 2E & EV2F), is a leucine in ubiquitin.

Points:

1. The authors need to demonstrate the functionality of Parkin Δ86-130 compared to full length Parkin, both by biochemical ubiquitylation assays in vitro, and by functional assays in a mitophagy model. They could also determine whether Parkin Δ86-130 C431S is still able to trap ubiquitin and be activated by pUb in a ubiquitylation assay to the same extent as WT Parkin.

We agree and now include Parkin Δ86-130 in the autoubiquitination assay (Figure 5F). The linker deletion mutant is activated by phosphorylation and shows autoubiquitination activity. Comparison with wild-type Parkin shows that the mutant has less activity possibly due to loss of Lys129 in the linker that serves as an autoubiquitination site (Sarraf et al., *Nature*, 2013). The shorter linker may also reduce ubiquitination of the Ubl domain that bears known ubiquitination sites at Lys27, Lys48 and Lys76 (Durcan et al. *EMBO J*, 2014; Sarraf et al., *Nature*, 2013). Nonetheless, the Δ86-130 mutant shows activity and all indications are that it faithfully reproduces the native autoinhibited structure of Parkin.

We note that superposition of our previous low-resolution structure of Parkin and Δ86-130 mutant confirms that a shorter linker doesn't lead to any major structural rearrangements (Figure EV1D). The major effect of the Δ86-130 deletion is that it allowed us to obtain a much higher resolution structure with the molecular details of the interface between the Ubl and RING1 domains.

2. *Figure 3C: Although pUb or the L266K mutation was able to increase Parkin S65 phosphorylation by PINK1, the difference is not significant, suggesting that the release of the UBL domain is not a prerequisite for S65 phosphorylation. The PINK1 kinase assay was a 30 min reaction, which may be too long, and therefore mask the importance of pUb binding in priming UBL phosphorylation. The authors should try a shorter time point, and use a pS65-specific antibody to quantify phospho-Parkin.*

We apologize for an error in the original manuscript. The experiment shown in the original Figure 3C was the result of a 5 min assay as stated in the Material and Methods and not 30 min as erroneously stated in the figure legend. We believe that the difference shown in that figure is significant but we have repeated the assays with less enzyme in order to better measure the initial rate of phosphorylation. We continued to use a gel containing the Phos-tag reagent to separate Parkin and phospho-Parkin since it allows us to compare the levels of phosphorylated and unmodified Parkin. This is a significant advantage over immunodetection, which only reports on the level of phosphorylated Parkin.

Figure 3C of the revised manuscript shows that WT Parkin is only phosphorylated in the presence of pUb whereas, under the same conditions, the L266K mutant was phosphorylated to the same extent in the presence or absence of pUb. We also included the pUb binding site mutant H302A Parkin in our assay and show that the addition of pUb to H302A Parkin doesn't promote its phosphorylation. We also show that the activation of Parkin phosphorylation by pUb is specific for the phosphorylated form of Ub. Addition of unphosphorylated Ub (mutated at Ser65 to prevent its phosphorylation) did not stimulate phosphorylation of Parkin by PINK1 (Figure EV3B).

Our data do not indicate that pUb binding is a prerequisite for Parkin phosphorylation; however, we see a consistent, significant, and large increase in Parkin phosphorylation upon release of the Ubl from either mutagenesis (L266K) or pUb binding.

3. *To identify the potential pUb binding site, the author hypothesized that the negatively-charged phosphate on S65 would bind to a positively charged patch on R0RBR, and thus they selected basic residues that were either located adjacent to the UBL domain (R271, R275, R314, R334, K369), or were bound to sulfates in the Δ86-130 structure (K161, K211, H302, R305). Do any other positively charged patches exist in regions outside of RING1+IBR? Mutation of H302 and R305 compromised pUb binding to R0RBR (Figure 4).*

There are positively charged patches outside of RING1+IBR such as the sulfate-binding site on RING0 (K161 and K211). Mutation of that patch did not impair pUb binding. We focused our study on positively charged residues located in regions where pUb was likely to bind based on ITC and NMR results showing competition between Ubl and pUb binding. Therefore it was plausible to think that the pUb-binding site would be located on the R0RBR module in proximity to the Ubl binding site (RING1-IBR region). Once we observed that mutations in the sulfate-binding site formed by H302 and R305 blocked pUb binding to Parkin (Figure 4A, 4B), we focused on characterizing that site. Our SAXS data and functional assays support H302-R305 as the principal site of pUb binding (Figure 4 D-F). It is possible that future studies will identify additional sites that interact with pUb or pUbl.

4. *Figure 5C: These data show that addition of pUb to R0RBR did not affect UbcH7 binding. This is an interesting observation, as R0RBR can bind to pUb with high affinity (Figure 3A). As the authors suggested, this experiment indicates that UBL domain has a positive role in Parkin E3 activity. The affinity of R0RBR+pUb and R0RBR+pUb+pUBL on UbcH7 binding (and Ub discharge) should be measured and included in Figure 5B. This point may be a significant contribution of this paper, since there are hints in the literature that UBL domain may not simply play a negative role in Parkin activity. For instance, the UBL domain has been shown to be essential for ubiquitin trapping in Parkin C431S (Zheng and Hunter, Cell Res 23:886). Moreover, the pUBL domain can stimulate*

the E3 activity of UBL-deleted Parkin (Kazlauskaitė et al. *Biochem J* 460:127) and is essential for Miro ubiquitylation (Kazlauskaitė et al. *Open Biol* 2:120080).

We thank the reviewer for the comment and we agree that a significant conclusion of our paper is that the Ubl plays the lead role in Parkin activation. We have carried out additional NMR titrations to confirm that Parkin phosphorylation increases UbcH7 binding more than deletion of the Ubl (Figure 5D) and we have expanded the autoubiquitination assays to show that pUb binding has only a small effect stimulating Parkin activity. With purified proteins, the main effect of pUb binding appears to be releasing the Ubl to enable its phosphorylation and to modestly increase E2 binding. In cells, pUb likely plays a more important role as the mitochondrial receptor for Parkin by virtue of its very high affinity.

The nature of the positive role of pUbl remains unclear. Other groups have shown that pUbl can activate RORBR *in trans* but we were not able to detect a specific interaction of pUbl. As mentioned above, a number of groups are pursuing the structure of the catalytically active Parkin. How pUbl contributes to Parkin activity is a topic of future investigation.

5. Figure 5D: In the pWT panel, it appears that addition of pUb did not further enhance the E3 activity of pParkin, apparently arguing against the importance of pUb. Why did pWT+pUb have lower E3 activity compared to WT+pUb, considering that WT Parkin has a low affinity for pUb and S65D ubiquitin did poorly in WT Parkin binding (Figure EV3)? To the reviewer, the results seems to support the idea that the H302A mutation makes the UBL phosphorylation essential for Parkin activity, with the two phosphorylation events being redundant rather than additive and cooperative. Here, there are some issues that the authors may take into consideration. First, does the Parkin autoubiquitylation activity really reflect the E3 activity toward its substrates? This could easily be tested. Second, does pUb inhibit Ub chain formation? *In vivo*, Parkin increases K63 chain formation on the damaged mitochondria, and because pS65 lies very close to K63, it may therefore affect the chain formation by Parkin. A C-terminal GG mutant form of pUb might be a better option. Moreover, a C431S mutant should be used as negative control to confirm that the autoubiquitylation activity occurs through a HECT-like mechanism.

To address the reviewer's concerns, we have repeated and expanded the autoubiquitination assays in Figure 5. The new assays tested additional mutations and used a shorter incubation time (45 min instead of 2 h) to better capture the initial rate of Parkin activity. In agreement with our UbcH7 binding experiments, the autoubiquitination assays demonstrate that Ubl phosphorylation is more important for stimulating Parkin activity than pUb binding. Addition of pUb to WT Parkin slightly increased autoubiquitination activity but Parkin phosphorylation had a much larger effect. We did not see synergistic activation (or inhibition) by addition of pUb to phosphorylated Parkin.

Importantly, the autoubiquitination assays show that the K273N, H302A, and R305A Parkin mutants are fully active when phosphorylated. We also show that the Δ 86-130 linker deletion mutant can be activated by phosphorylation although to a lesser extent than wild-type Parkin. The issue of chain specificity is interesting and will undoubtedly be examined in future work by ourselves and other groups.

The reviewer is correct in pointing out that autoubiquitination is not perfectly reflective of Parkin activity on physiological substrates. Nonetheless, we and others have shown that autoubiquitination is a useful proxy for monitoring the E3 activity of Parkin (Trempe et al. 2013, Chaugule et al. 2011). For example, the W403A REP mutation stimulates autoubiquitination which mirrors its faster to mitochondria in cells (Trempe et al. 2013). That study confirmed that autoubiquitination occurs through a HECT-like mechanism: the C431S mutant was complete inactive in autoubiquitination and UbcH7~Ub discharging assays.

Minor points:

1. If equimolar pUb can compete the binding of UBL domain to R0RBR (Figure 3B), why did S65D ubiquitin not pull down WT Parkin (Figure EV3A)?

According to our ITC experiments, S65D ubiquitin has a 200-fold lower affinity for Parkin than pUb, which explains the pull-down result. In light of our ITC, NMR and SAXS data with phospho-ubiquitin, the figure with the S65D phosphomimetic is obsolete and we have removed it from the manuscript as suggested by reviewer #2.

2. Page 6: Figure EV1D and Figure EV1E are mislabeled; there is no EV1E.

Thank you for pointing out this mislabeling. We have corrected the figure labeling in the text.

Referee #2:

Parkin is a key player in the mitochondrial quality control pathway and mutations in Parkin cause early-onset Parkinsons disease. Parkin has been shown to exist in an autoinhibited state that needs to be released to promote ligase activity and induce mitophagy. Release of autoinhibition is promoted by phosphorylation events induced by PINK1 that alter the overall structure of Parkin to allow interaction with its cognate E2 and ubiquitination of substrate proteins.

The manuscript by Sauve and colleagues describes the crystal structure of full-length Parkin from which a loop region has been deleted (delta 86-130) and provides an extensive biophysical characterization of Parkin and its interaction with phosphorylated ubiquitin which allow the authors to propose a location for the pUb binding site on Parkin. Furthermore, they explore the role of Parkin phosphorylation and pUb binding on the interaction with UbcH7 and based on their data present a model for Parkin activation by PINK1.

This is an interesting study that makes an important contribution to advancing our understanding of Parkin activation through phosphorylation of its Ubl domain and binding of pUb. It provides a first insight into the conformational changes occurring during the activation process which are likely to be highly dynamic and hence difficult to capture by crystallographic studies alone.

We would like to thank the reviewer for the many useful comments and suggestions. We have addressed most of the issues raised and carried out additional SAXS modeling to confirm the identification of the pUb-binding site.

Major concerns

- *The authors need to carry out ubiquitination assays to test if the crystallized fragment is indeed a good mimic of the wt protein or if the deleted loop may play an additional role that is not understood at the moment.*

Generally, the paper would profit from carrying out ubiquitination assays with mutants of those residues that have been identified to play an important role (eg L266 or H302) to test if their activity supports the function that has been ascribed to them. Similarly, it would be useful to link the ITC studies to ubiquitination assays to highlight how changes in affinities translate into changes in activity.

We thank the reviewer for the suggestion. We performed new autoubiquitination assays with the Δ 86-130 deletion mutant, the Ubl-binding site mutant N273K and the pUb-binding site mutants H302A and R305A (Figure 5F). The results show that all the mutants are active and strongly stimulated by Ubl phosphorylation.

As suggested by the reviewer, the new assays now allow us to link our ITC results with autoubiquitination. The ITC results in Figure 5B show that the phosphorylation of the Ubl is essential for efficient E2 binding and the autoubiquitination assays in Figure 5F show that phosphorylation similarly is required for high-level ubiquitination activity.

The linker deletion mutant is activated by phosphorylation and shows autoubiquitination activity but less than wild-type Parkin. This may be due to loss of Lys129 in the linker that serves as an autoubiquitination site (Sarraf et al., *Nature*, 2013). The Ubl also has three sites for autoubiquitination at Lys27, Lys48 and Lys76 (Durcan et al. *EMBO J*, 2014; Sarraf et al., *Nature*, 2013) and the shorter length of the Δ 86-130 linker may limit ubiquitination of the Ubl domain. Although beyond the scope of the current manuscript, we are planning to test the activity of the mutants more fully using Ub_{CH7}-Ub discharging assays and in cellular assays of Parkin activity.

- *Figure 2B: the comparison of spectra from the 1:1 and 1:2 pUbl + R0RBR mixtures clearly shows that there is a signal reduction in the 1:2 mixture, indicating that an interaction is taking place, though a weak one. Could the authors try to go to higher ratios and try to estimate a Kd for the interaction? Why do the authors show different regions for the HSQC spectra of phosphorylated and unphosphorylated Ubl domains?*

We agree with the reviewer that the NMR data show a weak interaction between pUbl and R0RBR. We believe that this is due to a residual interaction with the RING1 Ubl-binding domain. Phosphorylation of the Ubl domain decreases the affinity for binding to RING1 but does not completely inhibit it. Based on the change in affinity for pUb, it seems likely that the Ubl phosphorylation decreases the affinity for the RING1 by 10 to 20-fold. It would be interesting to have an independent measurement of the affinity but unfortunately we are limited in the concentrations of R0RBR accessible by NMR. The affinity of pUbl for R0RBR is an important question for future development.

The NMR spectra in Figure 2B and 2C have been replotted with identical axes.

- *The authors use the differences in Dmax values of wt full length Parkin compared to pParkin and the L266K mutant as an indication that their conformation changes and that the molecules have become elongated due to dissociation of the Ubl domain. However, in figure EV4E where the shape of R0RBR and its pUb complex are compared a similar difference in Dmax is regarded as "only slightly different". It's either one or the other. Please explain and rephrase.*

We thank the reviewer for pointing this out and acknowledge that we should have been clearer about this. The integral of the P(r) function is proportional to the forward scattering (I_0), which is proportional to the mass of the scattering particle (Putnam et al. *Q Rev Biophys*, 2007). Thus, the area under the curves for R0RBR and R0RBR+pUb are different because the first is 37kDa, and the second is 46kDa (Figure EV4C). If we rescale the two curves so that they have the same area under the curve, it is evident that they are very similar indeed (see graph below), and indeed the two data sets have identical Rg values (Figure 4C). By comparison, the area under the curves for all the full-length parkin variants are the same, but their curves look different, which is reflected in their higher Rg values (Figure EV2C,E). The D_{max} parameter is also poorly estimated, in part because it is used in the indirect Fourier transform to generate the P(r) function, and also because the P(r) curve is small in this region and contribute little to the overall scattering. Thus it is important to look not only at the actual D_{max} value, but mostly at how steeply the P(r) functions decreases at higher r values: for example, it is quite obvious that the P(r) at $r=100$ is higher for L266K and pParkin, compared to WT Parkin (Figure EV2E). We have rephrased the text corresponding to these sections to be more precise about differences and similarities in the SAXS data.

- The pull down assays shown in figure 3EV3A are not very convincing: The ITC data show that pParkin binds to pUb with an affinity of 17nM but there's hardly any interaction seen in the pull down between S65E Parkin and S65D ubiquitin. Does this mean that the mutants are not very good mimetics of the phospho forms? There's a ~15 fold difference in affinities between S56E Parkin and pParkin for pUb, but the complex should be still tight enough to be pulled down together. Why did the authors do the pull downs with the phosphomimetics whereas the ITC experiments were done with the phospho forms? I am also surprised that a complex that has an affinity of ~0.4 μ M (wt Parkin + pUb/S65D Ub) cannot be detected in a pull down assay. Overall, I fail to see what the pull down experiments add to the extensive characterization by ITC and they should be removed unless the authors think they add more information. If this is the case please discuss.

We agree with the reviewer that this figure doesn't add much to the paper and we have thus decided to take it out. This figure was historically important for us, because this was our first demonstration that phospho-mimetic S65D-Ub binds parkin and competes with the Ubl binding. According to ITC results (not shown here), S65D ubiquitin has a 200-fold lower affinity for Parkin than pUb. This explains this pull-down result. The figure is now obsolete given the ITC data obtained with the phosphorylated proteins.

- I'm not convinced by the approach taken to derive the model presented in figure 4D and figure 4C isn't very meaningful either. It would be much more useful if the authors modelled a molecular envelope for the RORBR/pUb complex based on their experimental data and then fitted the RORBR structure into this envelope. This should indicate where the binding site for pUb is - ideally close to H302.

Please add the scattering data for RORBR+Ub to figure EV4C.

We thank the reviewer for the suggestions. 3D models derived from 1D SAXS data are underdetermined. Thus, any constraints that can be added to reduce the number of fitting parameters increase confidence in the 3D model. This was the basis for our approach: we used currently available atomic models for RORBR and pUb and rigid-body docking combined with mutagenesis-based distance constraints to model the SAXS data, with very satisfactory fit. All we are saying is that the SAXS-derived docking model is consistent with our data. We have thus decided to maintain the docking results in our manuscript.

Nonetheless, we have followed the reviewer's suggestion and performed ab initio shape calculation to test if it would be possible to locate the pUb-binding site

independently of the mutagenesis results. To validate the approach, we have performed shape determination using the R0RBR data only, which we know fits well to the crystal structure. The resulting envelope fitted well to chain A from the R0RBR crystal structure (Figure EV4D), although evidently some of the loops in RING0 and IBR “stick out” of the envelope: GASBOR tries to maximize the number of close-distance neighbors for each dummy atom, thus effectively smoothing the molecular envelope. Moreover, the averaging process would get rid of such singularities. With these limitations in mind, we pursued shape determination of the R0RBR:pUb complex. Superposition of the R0RBR structure into the envelope revealed unaccounted “density” between RING1 and RING0 (Figure EV4E), which could accommodate pUb bound to His302 (Figure 4D). While the fit is not perfect, notably in the region of the IBR domain that clearly sticks out of the envelope, it confirms our identification of the binding site. We have included this new result in the re-submitted manuscript.

The scattering data for R0RBR+Ub and fit from the rigid body modeling are shown in Fig. EV4G.

Minor concerns

- A 2.5 Å crystal structure is not "high-resolution".

We agree. We now write “higher resolution” to indicate that the resolution is higher than our previous 6.5 Å structure.

- page 6: Please add an explanatory sentence to the main text explaining how SAXS experiments exclude model 2 as the figure and associated text are only shown in supplementary material.

We changed the text and reported Chi2 values showing that model 1 fits the SAXS data much better than model 2.

- How often have the ITC titrations been repeated and how have the error estimates given been calculated?

Most of the ITC titrations were done with different concentrations of proteins, but we only report the values of the curves with the most optimized fit. The reported errors reflect the fitting of the curve (and therefore the Kd, N, ΔH estimations) with the data points.

- page 6: Where is figure EV1E? Should this read EV1D?

We apologize for this mislabeling. There was no Figure EV1E in the original manuscript. We have corrected the text.

- It would be helpful for the reader to highlight the E2-binding site in figure 1B.

We have highlighted it in Figure 1B and a model with Parkin and UbcH7 in Figure 5A.

- page 7: The authors should add a figure (could be supplementary) to illustrate the point made about the flexibility of the IBR domain and the different positions it adopts. Do the authors think this is physiologically relevant?

We have added Figure EV1E to show the range of IBR orientations observed in the different crystal structures. The flexibility of the IBR might be physiologically important: there is a Parkinson mutation in the hinge that connects RING1 to IBR (G328E). That mutation abrogates autoubiquitination activity (Trempe et al. 2013), suggesting that the flexibility of the IBR is important for its activity.

-What is the point the authors want to make with figure EV2A? These titrations have already been shown in figure 2.

The idea was to show the ITC results of the middle and right panels of figure 2A on a scale that allow a better appreciation of the data points and the curves and to show the ITC result of the left panel of figure 2A done at a protein concentration even lower than for the two other panels. Even at lower concentration, the data points of Ubl with R0RBR can be fitted with a curve giving a Kd estimate comparable to that obtained at the higher concentration (in Figure 2A). No curve could be fitted with confidence for the pUbl with R0RBR and Ubl with L266K Parkin.

And why did the authors feel they needed to confirm data from ITC and NMR experiments with a pull down assay, especially one that is not a very good one (figure 2EV2B)? This assay doesn't add anything and should be removed.

We believe that converging evidence from multiple experimental designs increase confidence in the results. The distortion in the gel and the large bands come from the protein overload, which was essential in this pull-down because the interaction with the Ubl is relatively weak (16 uM) and leads to rapid dissociation. Nonetheless, the gel has all the appropriate controls and it is quite clear that phosphorylation of the Ubl dramatically reduces binding to R0RBR. As it is not in the main figure, we propose to leave the gel as a supporting figure.

- bottom page 8: Please say explicitly how SAXS provides shape information through the calculation of the pair-distance distribution so that a non-expert can follow the argument.

We have added a sentence explaining how the SAXS pair-distance distribution function provides information about interatomic distances in a protein, which can be used to determine whether intramolecular interactions are disrupted.

- page 9: Please add a sentence explaining what is meant by "an alternative conformation of the Ubl domain". Please also say explicitly that the perturbations were mapped onto the Ubl domain of the.

We have added a sentence in the text to clarify what we meant by “alternative conformation”, and mentioned that the perturbations were mapped onto the Ubl domain of the $\Delta 86-130$ Parkin structure.

- Figure 3B: I am slightly surprised that the addition of 1 equivalent R0RBR to a 100 uM solution of Ubl leads to an almost complete loss of signal given that their affinity is only 22 uM. Are the authors sure that the concentrations reported are correct?

The loss of signal is not caused a reduction in the amount of free ^{15}N -Ubl but by NMR signal relaxation through the complex. The magnetization relaxation rate of the complex is much faster than that of free Ubl domain so that a small amount of the complex has a large effect on the observed line broadening and results in signal loss. Technically, we are observing transferred relaxation rates under conditions of fast exchange between the free and bound forms.

The NMR signal loss we observe is useful for qualitative assessment of binding but the loss depends on a multitude of factors such as exchange rates, chemical shifts, etc. Without knowing the relaxation rates for the complex, it is impossible to calculate the amounts of free and complexed UbcH7 based on the amount of signal loss. However, under the assumption that related complexes have similar relaxation rates, we are able to make comparisons between molecules. For example, the absence of signal loss of phosphorylated ^{15}N -Ubl in the presence of R0RBR undoubtedly indicates weak binding (Figure 2C). Similarly, the reappearance of the ^{15}N -Ubl signals upon addition of pUb is strong evidence of competitive binding (Figure 3B).

The concentrations of the proteins are correct. At 100 μM concentration of both proteins and a K_d of 16 μM , the expected amount of free Ubl should be about 30 μM . However, due to fast NMR relaxation of the complex, essentially all of the Ubl signals are lost.

- Please show all the ITC titrations that have been combined in a given figure on the same scale for easier comparison.

We understand the reviewer's concern and attempted to show the ITC curves on the same scale. However, for the reasons explained below, we decided to maintain the ITC titrations in Figures EV3A and EV5A on different scales for a better representation of the curve fitting. ITC experiments were performed at different concentrations as indicated on each thermogram. The protein concentration for the titrations were chosen to optimize the C-value ($C = \{[M]_{\text{tot}} / K_d\} * N$, where $[M]_{\text{tot}}$ is the concentration of the macromolecule (here protein) in the cell and N is the ligand:macromolecule stoichiometry of binding) for a more accurate estimate of the K_d (see graphic below). Since the heat change signal is proportional to the protein concentration, the signals from titrations performed at lower concentration would be very small if plotted on the same scale as those from titrations performed at higher protein concentration. As a consequence, the experiments done at the lower concentrations to reduce the C-value and therefore increase the accuracy of the K_d determination were those with the highest binding affinity.

- A cartoon explaining the conformational changes and model described in the discussion would be very helpful.

Thank you for the suggestion. We have now added an extra figure (Figure 6) to complement our discussion.

Referee #3:

This manuscript integrates multiple approaches to provide fundamental information on how parkin activity is regulated by phosphorylation of the Ubl domain and of ubiquitin. A 2.54Å resolution crystal structure of full length parkin with a linker region deleted is solved and SAXS, NMR, ITC and site-directed mutagenesis used to characterize the effects of Ubl and ubiquitin phosphorylation. This reviewer is enthusiastic toward this manuscript, which is compelling with multiple elements that weave together in an elegant way. Some technical concerns however are listed below that should be addressed, although can probably be without further experiments.

1. The authors should explicitly state whether the NMR data are collected and processed in an identical manner when being compared to each other. For example, in Fig 2C, it's probably true that the sample concentration, experimental parameters (especially number of scans), and processing parameters are identical for the left and center panel, but this is not stated.

We agree with the reviewer that this information is crucial to interpret the data and we have remedied by specifying those experimental parameters. As indicated in the figure legends and Materials & Methods, the spectra were scaled to account for the number of scans and differences in concentration of the labeled protein.

We did not observe strong effects of experimental parameters (pulse sequence, probe tuning etc) in the qualitative analysis of binding. We observed no major differences in the interpretation of duplicate titrations performed at two different field strengths. For the most accurate comparisons, the experiments were done under identical conditions on the same spectrometer on the same day.

2. It seems that all NMR experiments are done at pH 7.4. This is not ideal for observing amide groups because of exchange with water. Moreover, is the pH reported for buffer or final sample? As stated above, the authors should explicitly state that the pH is not changed in comparisons in Materials and Methods if this is the case. Confidence would be higher if 1D traces were included showing broadened lineshapes rather than just disappearance of signals. A couple of traces could be shown as a figure insert even.

The reviewer is correct that acidic pH is preferable for HN-HSQC because of slower amide exchange rates. We chose pH 7.4 as a compromise because both pUbl and Parkin are more soluble and stable above pH 7.0. If solvent-driven amide exchange were the major contributor to signal loss, then we would see less loss from residues at the intermolecular interface. In tightly bound complexes, those residues would be protected from solvent exchange. Instead, we observed that those residues showed the greatest signal loss. Overall, we saw no evidence that signal loss was related to solvent accessibility.

The pH is reported for the buffer. All samples were buffer exchanged by gel filtration in the same manner and the final sample pH should be the same in all cases. This is now stated in the Materials and Methods.

To aid in the interpretation of the UbcH7 binding studies, we have added panels with peak heights for selected resonances as a function of protein concentration (Figure 5D and EV5C) and for different mutants at a fixed concentration (Figure 5E). Plotting the decrease in peak heights should facilitate the interpretation of the NMR binding studies.

We do not see marked differences in the amide proton line widths in the titrations. The majority of signal loss likely occurs during the delays for heteronuclear magnetization in the HSQC spectra. As described above, uncertainties in the relaxation rates of the bound state limit our ability to quantitatively analyze the titrations with different complexes. Even so, the large and reproducible differences in peak intensities can be used to compare the binding affinities of UbcH7 for different Parkin mutants and complexes. For interactions that are high enough to be measured by ITC, we observed perfect agreement between the NMR and ITC data.

3. Regions of the 15N NOESY should be displayed to support the authors' conclusion of no changes in beta-strand alignment following Ubl phosphorylation.

We have added a new Figure EV2H that shows similar cross-strand NOEs in samples of Ubl and pUbl. We do not see any evidence of a beta-sheet strand shift as observed in the minor conformation of pUb by Wauer & Komander.

4. The HSQC spectra corresponding to the chemical shift perturbation for pUbl vs Ubl should be included.

The HSQC spectra are now included in Figure EV2G. A plot of chemical shift perturbations by residue is presented in Figure EV2F.

5. The authors need to address further what the changes to NMR signals from Ubl phosphorylation reflect. Clearly they do not think that these changes are from secondary structure changes based on the NOESY spectrum. The authors write however that they "are consistent with structural changes," thus potentially conveying a mixed message. Amides are highly sensitive to their chemical environment and can shift by changes in dynamics, reconfiguration of interactions with neighboring amino acids, etc, as the authors probably know but their readers may not. The authors should indicate what their data collectively best reflects.

The reviewer correctly points out that our text did not accurately reflect our results. The chemical shift perturbations induced by the phosphorylation of the Ubl are probably caused by side chain reconfigurations, especially near the phosphorylation site Ser65 that experiences the largest perturbations. We have modified the text to better explain the significance of the chemical shift changes and our data.

2nd Editorial Decision

23 July 2015

Thank you for submitting the revised version. Your manuscript has now been re-reviewed by referee #2 and as you can see below the referee appreciates the introduced changes. I am therefore very pleased to accept the manuscript for publication here.

REFEREE REPORT

Referee #2:

The authors have done a good job addressing my initial concerns about the manuscript. This is an interesting study that makes an important contribution to our understanding of Parkin activation.